# Conversational Time Series Foundation Models: Towards Explainable and Effective Forecasting

## Abstract

The proliferation of time series foundation models has created a landscape where no single method achieves consistent superiority, framing the central challenge not as finding the best model, but as orchestrating an optimal ensemble with interpretability. While Large Language Models (LLMs) offer powerful reasoning capabilities, their direct application to time series forecasting has proven ineffective. We address this gap by repositioning the LLM as an intelligent judge that evaluates, explains, and strategically coordinates an ensemble of foundation models. To overcome the LLM's inherent lack of domain-specific knowledge on time series, we introduce an R1-style finetuning process, guided by SHAP-based faithfulness scores, which teaches the model to interpret ensemble weights as meaningful causal statements about temporal dynamics. The trained agent then engages in iterative, multi-turn conversations to perform forward-looking assessments, provide causally-grounded explanations for its weighting decisions, and adaptively refine the optimization strategy. Validated on the GIFT-Eval benchmark on 23 datasets across 97 settings, our approach significantly outperforms leading time series foundation models on both CRPS and MASE metrics, establishing new state-of-the-art results.

## 1 Introduction

Large Language Models (LLMs) have demonstrated remarkable capabilities in reasoning (Wei et al., 2022; Højer et al., 2025), decision-making (Schmied et al., 2025), and contextual understanding (Berglund et al., 2023), making them natural candidates to address the fundamental challenges in time series forecasting (Fan et al., 2025). The field has witnessed an unprecedented proliferation of sophisticated models, including Transformers (Wen et al., 2022), N-BEATS (Oreshkin et al., 2019), and foundation models (Das et al., 2024; Faw et al., 2025a). Yet, no single model achieves consistent superiority across the diverse landscape of real-world forecasting scenarios . In addition, time series forecasting is fundamentally a reasoning-centric task that transcends mere pattern recognition (Luo et al., 2025a). Given this dual requirement and the heterogeneous capabilities of modern models, the critical question shifts from which model performs best to *how do we orchestrate the convergence of algorithmic optimization and human understanding, transforming opaque numerical forecasting into transparent reasoning that practitioners can validate, trust, and act upon?*

When directly applied to time series forecasting, LLMs face limitations (Li & Pedersen, 2025; Merrill et al., 2024; Tang et al., 2025b; Tan et al., 2024). While they can understand temporal patterns conceptually (e.g., "demand spikes during holidays" or "policy changes trigger regime shifts"), their training on language rather than numerical regression prevents them from reliably generating precise quantitative predictions. Recent attempts to address these limitations have taken several forms: direct prediction methods to update LLMs to perform regression tasks they were not designed for, resulting in computationally expensive processes consistently outperformed by traditional models (Bumb et al., 2025; Cao et al., 2024a; Jin et al., 2024). Multi-modal alignment approaches primarily use LLMs as embedding generators, underutilizing their so-

phisticated reasoning capabilities (Xie et al., 2024; Zheng et al., 2025). Supervised finetuning (SFT) or reinforcement learning finetuning struggle with the fundamental mismatch between linguistic reasoning and numerical precision, requiring extensive computation for marginal improvements (Liu et al., 2025c; Wang et al., 2025b). Zero-shot agent approaches preserve general reasoning but lack the domain-specific understanding needed for informed model selection (Gruver et al., 2023a; Ye et al., 2024). These paradigms overlook LLMs' core strengths, which, instead of leveraging their exceptional **capacity for high-level reasoning**, either force numerical computation or treat LLMs as passive components.

To address this gap, we propose TSOrchestr, which positions LLMs as intelligent judges that evaluate, explain, and strategically coordinate ensembles of specialized forecasting models, thereby combining the numerical precision of time series foundation methods with the interpretive power of language models. For LLMs to function as effective decision makers, they need multiple strategic options to evaluate and choose from, as a single model would reduce the LLM to a mere pass-through layer without meaningful contribution. By orchestrating with multiple models, we create a rich decision landscape where the LLM must evaluate trade-offs and make strategic choices, for example, reasoning about whether to prioritize $A$ model's high-frequency sensitivity versus $B$'s long-range pattern recognition given current conditions. However, this decision-making role exposes a critical knowledge gap that LLMs inherently lack in understanding what ensemble weights represent in time series contexts. Without model updating, an LLM cannot comprehend that a $0.7/0.3$ weight split reflects judgments about seasonal stability versus pattern complexity, or that weight adjustments represent fundamental reinterpretations of temporal dynamics. R1-style finetuning (Guo et al., 2025) becomes essential to teach the LLM this specialized vocabulary, which trains it to understand that weights encode specific hypotheses about which temporal patterns dominate and which architectures best capture them. Through finetuning guided by SHapley Additive exPlanations (SHAP)-based faithfulness scores, the LLM learns to interpret weight assignments as causal statements ("high weight on model A indicates strong autoregressive patterns"), transforming abstract numerical weights into semantically meaningful decisions that can be reasoned about, explained, and strategically adjusted.

Our finetuned reasoning agent transforms static ensemble optimization into a dynamic, adaptive system. The agent first performs a forward-looking assessment, analyzing historical performance patterns to determine if current weight combinations will remain valid and to identify potential regime shifts. Second, the agent engages in iterative, multi-turn conversations to refine the optimization by analyzing performance and suggesting updated weight configurations. Finally, it provides causally-grounded explanations by aligning its reasoning with SHAP-measured contributions, articulating why specific models receive higher weights based on their impact on temporal components like trend and seasonality, culminating in interpretability reflection. We evaluated our approach on the GIFT-Eval (Aksu et al., 2024) benchmark, which comprises 23 time series datasets with 97 settings to ensure generalizability. Our model significantly outperformed the best time series foundation models on both CRPS and MASE metrics, establishing new state-of-the-art results across diverse forecasting tasks.

## 2 RELATED WORK

**LLM/Agentic-Based Time Series Forecasting.** Recent approaches leverage LLMs for time series forecasting through various paradigms, yet all fundamentally misalign with LLMs' text-based capabilities. Direct prediction methods force LLMs to generate numerical outputs: LLMTIME (Gruver et al., 2023b) treats forecasting as next-token generation, TEMPO (Cao et al., 2024b) decomposes series for component prediction, DP-GPT4MTS (Liu et al., 2025a) uses dual prompting, and NNCL-TLLM (Bogahawatte et al., 2024) learns time-series-compatible prototypes. Context-enhanced approaches incorporate external information—CiK (Williams et al., 2025) shows gains with textual context but catastrophic failures in direct prediction, Hybrid-MMF (Kim et al., 2024) jointly forecasts numbers and narratives with negative fusion results, CMLLM (Zhu et al., 2025) converts SCADA signals to text, and Tang et al. (2025a) reprograms histories into prose.

Retrieval-augmented methods enhance predictions with historical patterns: RAF (Tire et al., 2024) builds dataset-specific databases, TimeRAG (Yang et al., 2024) slices and reprograms segments, TimeReasoner (Wang et al., 2025d) implements structured reasoning, and Time-R1 (Luo et al., 2025b) aligns scripted chains with supervised learning. Multi-agent systems introduce ensemble thinking: Zhang et al. (2025) uses competitive hypothesis pruning, NewsForecast (Wang et al., 2024) employs critique-revise loops, TimeXL (Jiang et al., 2025) iterates prediction-reflection-refinement, CAPTime (Yao et al., 2025) routes among probabilistic experts, CoLLM (Wang et al., 2025c) routes between models using confidence scores, and DCATS (Yeh et al., 2025) uses proposal-evaluate-refine cycles. Despite these advances, all existing approaches position LLMs as direct numerical predictors rather than leveraging their reasoning strength. Our framework fundamentally differs by positioning LLMs as intelligent judges orchestrating specialized forecasting models, separating high-level reasoning from numerical computation for superior, interpretable performance.

**Time Series Foundation models.** Recent foundation models for time series have achieved impressive scale and zero-shot capabilities, yet remain fundamentally limited as black-box predictors(Liu et al., 2024; Ansari et al., 2024). TimesFM (Faw et al., 2025b) operates on patches of time series data using a Transformer architecture with continued pre-training for zero-shot and few-shot forecasting. Toto (Cohen et al., 2024), trained by Datadog on nearly a trillion data points from observability data, demonstrates strong zero-shot prediction on unseen series. TabPFN-TS (Hoo et al., 2025), despite having only 11 million parameters trained purely on synthetic tabular data. Sundial (Liu et al., 2025b) scales to a trillion time points for adaptability across benchmarks. Timer casts forecasting as generative sequence modeling to leverage LLM capabilities. While these models achieve impressive performance, they operate as monolithic predictors without interpretable reasoning for their predictions, which is a gap our LLM-as-judge framework addresses by orchestrating multiple specialized models with explainable selection and weighting decisions.

**Beyond Forecasting: LLMs for Time Series Tasks.** LLMs have been applied to various time series tasks beyond forecasting. For classification, HiTime (Tao et al., 2024) aligns time series with textual semantics to generate class labels, while FinSrag (Xiao et al., 2025) predicts stock movements through retrieval and prompting. For anomaly detection, Zhou & Yu (2025) finds image inputs often outperform text for identifying anomalies. MedTsLLM (Chan et al., 2024) handles segmentation by projecting LLM outputs to masks and boundaries. ChatTime (Wang et al., 2025a) unifies multiple tasks by expanding tokenizers with discretized value symbols. These diverse applications demonstrate LLMs' versatility in time series analysis, yet none leverage their reasoning capabilities for model orchestration as we propose.

## 3 METHODOLOGY

### 3.1 PROBLEM FORMULATION

Given historical time series $\mathbf{y}_{1:t} \in \mathbb{R}^t$ and $M$ candidate forecasting foundation models $\mathcal{M} = \{m_1, m_2, ..., m_M\}$, our system produces both predictions and explanations through:

$$\hat{\mathbf{y}}_{t+1:t+h}, \mathcal{A} = f(\mathbf{y}_{1:t}, \mathcal{M}, \theta_{ensemble}, \theta_{LLM}) \tag{1}$$

where $\hat{\mathbf{y}}_{t+1:t+h} \in \mathbb{R}^h$ is the forecasting results for horizon $h$, $\mathcal{A}$ is the human-readable explanation of model weights and reasoning, $\theta_{ensemble}$ represents the SLSQP optimizer that determines weights $\mathbf{w}^*$, $\theta_{LLM}$ is the reasoning agent that represents the LLM-based agents responsible for both decision-making for continuing optimization or accept current results and interpretability reflection for generating human-readable output.

### 3.2 WEIGHT OPTIMIZATION

Real-world time series rarely maintain stationary distributions, such as economic regimes shift, seasonal patterns evolve, and external events create structural breaks. This temporal heterogeneity poses a fundamental challenge: no single model can optimally capture all regime-specific patterns simultaneously. We formalize

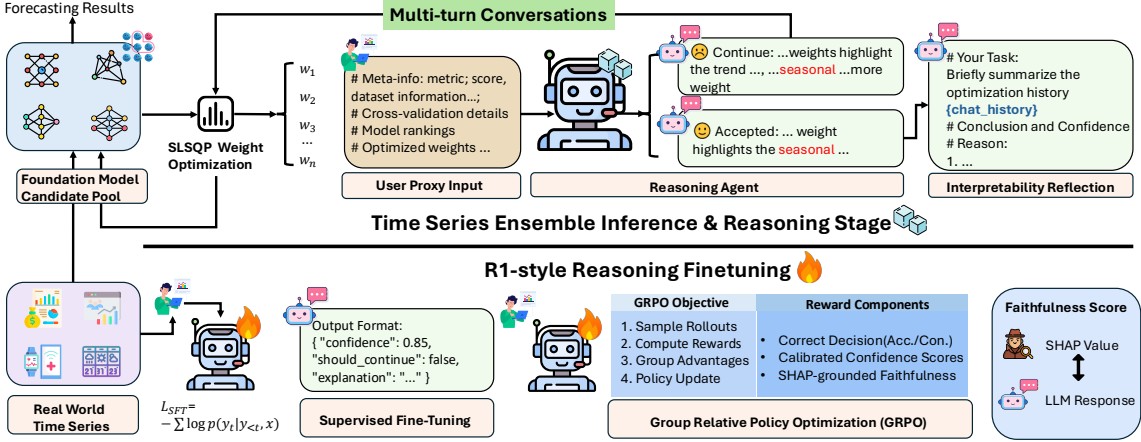

Figure 1: Model Architecture. The top stage shows an LLM agent guiding SLSQP ensemble optimization through iterative reasoning. The bottom stage details the agent's R1-style finetuning, where a SHAP-aligned Faithfulness Score rewards causally-grounded explanations during GRPO training.

this limitation and demonstrate how ensemble methods naturally overcome it. First, we quantify the inherent difficulty through the *Temporal Incompatibility Index*, $I_T(\mathcal{D})$, defined in Eq.13, which measures the degree to which similar historical patterns lead to divergent future outcomes.

**Theorem 3.1** (Ensemble Superiority Time Series forecasting). *For time series with temporal incompatibility $I_T(\mathcal{D}) > 0$, given the model diversity $M$, the optimal ensemble achieves:*

$$L^*_{ensemble}(\mathcal{D}) \leq L^*_{monolithic}(\mathcal{D}) - \Omega(I_T(\mathcal{D}), M) \tag{2}$$

*where $\Omega(I_T, M) = \frac{I_T \cdot \log M}{1 + \exp(-\kappa \cdot I_T)}$ quantifies the ensemble advantage as a function of incompatibility.*

This theorem (proof in Appendix D) establishes that ensembles have a theoretical advantage proportional to both the data's temporal complexity and the diversity of constituent models. In practice, each foundation model naturally specializes in different regime-specific patterns, and their weighted combination can approximate the true multi-regime dynamics that no single model can capture. Therefore, we employ Sequential Least Squares Programming (SLSQP)(Joshy & Hwang, 2024) to discover optimal ensemble weights that exploit these specialization patterns. We solve the constrained optimization problem: $\mathbf{w}^* = \arg\min_{\mathbf{w}} L(\mathbf{y}, \mathbf{Xw})$ s.t. $\mathbf{w} \geq 0$, $\sum_{i=1}^{M} w_i = 1$, where $\mathbf{y} \in \mathbb{R}^N$ represents true values, $\mathbf{X} \in \mathbb{R}^{N \times M}$ contains individual model forecasts, and $L$ is the chosen forecasting metric (e.g., MAE, MASE). The non-negativity and sum-to-one constraints ensure a valid convex combination. Crucially, the optimization automatically discovers weights that maximize coverage of the temporal complexity landscape. Models that excel in specific regimes receive higher weights for metrics where their specialization matters, while generalist models provide baseline coverage. This implicit regime discovery emerges naturally from the optimization process without requiring explicit regime identification.

However, the SLSQP optimization in Eq. 3.2 discovers weights $\mathbf{w}^*$ that maximize the ensemble advantage $\Omega(I_T, M)$ for the observed incompatibility level, but assumes this level remains stationary. Yet real-world time series exhibit dynamic incompatibility where $I_T$ itself fluctuates—periods of stable trends (low $I_T$) alternate with regime transitions (high $I_T$), and the frequency and intensity of these shifts vary over time.

When $I_T$ increases unexpectedly, the static weights cannot redistribute to maintain the theoretical ensemble advantage, precisely when adaptation matters most.

This limitation motivates our Agent-guided refinement (Section 3.3), where the LLM reasons about future $I_T$ evolution by analyzing patterns in the audit trail $\mathcal{A}$. Through the accumulated optimization history, the LLM identifies recurring incompatibility patterns. By extending these learned patterns forward and incorporating contextual signals, the LLM anticipates whether upcoming periods will exhibit higher or lower temporal incompatibility. This enables preemptive weight adjustments:

$$w_{\text{refined}} = w_{\text{SLSQP}}^* + \Delta w_{\text{LLM}} \tag{3}$$

where $\Delta w_{\text{LLM}}$ represents adjustments based on forward-looking regime analysis. The synthesis transforms static optimization into adaptive weighting that maintains the ensemble's theoretical advantage $\Omega(I_T, M)$ even as incompatibility evolves beyond historical patterns.

### 3.3 AGENT SYSTEM

#### 3.3.1 THE REASONING AGENT AS METACOGNITIVE CONTROLLER

We formulate the weight optimization as a reasoning problem where an LLM agent acts as a metacognitive controller over the mathematical optimization process. Unlike traditional approaches that blindly accept cross-validation results, our agent reasons about the gap between historical validation windows and future forecasting regimes.

The agent orchestrates iterative refinement through two capabilities: (i) strategic exploration by iteratively selecting metrics $L_k \in \mathcal{L}$ to test specific hypotheses about weight quality from optimizing Eq. 3.2 , (ii) evaluative reasoning that synthesizes evidence into confidence scores, decision rationales, and next hypotheses:

$$(\mathcal{C}_k, \mathcal{D}_k, L_{k+1}) = \text{LLM}_{\text{eval}}\{\text{Align}(\mathbf{w}_k^*, \mathcal{P}_{cv}), \text{Match}(\mathbf{w}_k^*, \mathcal{F}_{data}), \text{Future}(\mathbf{w}_k^*, \mathcal{K}_{domain})\} \tag{4}$$

where $\mathcal{C}_k \in [0, 1]$ is the confidence score, $\mathcal{D}_k \in \{\text{continue}, \text{accept}\}$ is the decision for the current round of optimization, $L_{k+1} \in \mathcal{L}$ is the recommended next metric, $\mathcal{P}_{cv}$ represents cross-validation performance across all metrics in $\mathcal{L}$, $\mathcal{F}_{data}$ contains dataset characteristics, and $\mathcal{K}_{domain}$ encodes domain knowledge such as model capabilities. The LLM evaluates weights across three dimensions: *Align* ensures performance-weight coherence and mathematical soundness, *Match* verifies dataset-model compatibility and complementarity, and *Future* assesses generalization risks and ensemble robustness.

The agent engages in multi-turn conversations with a user proxy to thoroughly evaluate each weight configuration. This dialogue structure allows the agent to: (1) articulate initial hypotheses about weight quality, (2) respond to challenges about potential weaknesses, and (3) refine its confidence assessment through iterative questioning. The final weight selection occurs when the LLM determines with high confidence that further optimization offers diminishing returns: $\mathbf{w}^* = \arg\max_{\mathbf{w} \in \mathcal{W}_{explored}} \mathcal{C}_{LLM}(\mathbf{w})$, where $\mathcal{C}_{LLM}$ represents the LLM's learned confidence function. This transforms optimization from a black-box numerical process into an interpretable reasoning task where the LLM can explain why specific weights represent the optimal trade-off between performance and robustness.

### 3.4 R1-STYLE FINE-TUNING FOR REASONING AND DECISION CAPABILITY

While LLMs can analyze optimization trajectories and reason about future temporal incompatibility evolution, they lack the domain-specific intuitions critical for ensemble weight selection. Time series optimization demands recognizing numerical convergence patterns, calibrating stopping decisions against overfitting risks, and understanding how validation performance translates to future forecasting accuracy—capabilities absent from language pretraining. Raw LLMs exhibit problematic behaviors: terminating prematurely on

superficial improvements, continuing indefinitely without recognizing diminishing returns, or failing to distinguish genuine robustness from validation artifacts. We address these gaps through a two-stage training approach that transforms a general-purpose LLM into a specialized ensemble-decider: supervised fine-tuning (SFT) teaches structured decision protocols through expert-annotated trajectories, while reinforcement learning (GRPO) refines these capabilities by directly optimizing for held-out forecasting performance, developing the mathematical intuitions that bridge linguistic reasoning and numerical optimization expertise.

### 3.4.1 SUPERVISED FINE-TUNING (SFT)

While LLMs can analyze patterns, they lack inherent understanding of optimization dynamics—when marginal improvements justify continuation versus when convergence indicates termination. Simply prompting LLMs yields inconsistent decisions: they may terminate prematurely at local improvements or continue indefinitely chasing negligible gains. We need to teach the model what constitutes meaningful progress in ensemble optimization and how to recognize genuine convergence patterns. Real optimization trajectories alone provide insufficient coverage of decision boundaries. Most trajectories contain obvious decisions (clear improvement or convergence), leaving the model undertrained on marginal cases where $\Delta = L_{\text{current}} - L_{\text{best}}$ approaches zero. We therefore construct a mixed dataset combining real trajectories with synthetically augmented boundary cases, ensuring the model learns both authentic optimization dynamics and robust decision calibration across the full spectrum of scenarios.

We implement SFT by optimizing next-token prediction loss on structured instruction-response pairs:

$$\mathcal{L}_{\text{SFT}} = -\sum_{t=1}^{T} \log p_\theta(y_t | y_{<t}, x) \tag{5}$$

where $x$ encodes the optimization state (metrics, weights, history) and $y$ contains both internal reasoning and final JSON decision. Our dataset stratifies decisions into three categories: clear-cut cases ($|\Delta| > 0.01$) establishing basic decision logic, marginal cases ($0.001 < |\Delta| \leq 0.01$) teaching boundary calibration, and ambiguous scenarios requiring weight distribution analysis beyond simple metrics. The reasoning-before-decision format forces explicit analysis, reducing hallucination while maintaining interpretability. Through this structured training, the model learns to follow multi-step reasoning protocols, ground decisions in quantitative evidence, and generate consistent outputs—providing the foundation for subsequent RL refinement, where it learns to optimize for actual forecasting performance.

### 3.4.2 REINFORCEMENT LEARNING (RL) WITH GRPO

SFT only learns to imitate expert labels without understanding the downstream impact on forecasting performance. The model may confidently suggest termination when further optimization could yield meaningful improvements, or recommend continuation despite genuine convergence. Moreover, LLMs can generate plausible-sounding explanations that misrepresent the actual causal factors driving ensemble performance—a critical issue when decisions affect real forecasting systems. We need the model to optimize for actual forecasting outcomes while ensuring its reasoning faithfully reflects the true optimization dynamics.

**Faithfulness-Aware Reward Design.** Standard RL rewards focusing solely on decision accuracy fail to address explanation quality. A model could learn to make correct decisions through spurious correlations while providing misleading justifications. We therefore introduce a faithfulness score that measures alignment between LLM explanations and ground-truth causal effects computed via SHAP (Shapley Additive exPlanations) analysis. Specifically, we employ SHAP analysis to create counterfactual time series by intervening on specific temporal concepts $\mathcal{C} = \{\text{Trend}, \text{Seasonality}, \text{Residual}\}$. For each subset $S \subseteq \mathcal{C}$, we evaluate model performance $f(S)$, where the presence or absence of components represents causal interventions, which can be calculated in Eq. 18 by introducing $do$ operator (Pearl, 1995). The causal faithfulness

score measures alignment between SHAP-derived causal effects and LLM explanations:

$$r_{\text{faith}} = \text{PCC}(\text{CE}(x, \mathcal{C}), \text{EE}(x, \mathcal{C})) \tag{6}$$

where $\text{CE}(\cdot)$ is the vector of SHAP-derived causal effects, $\text{EE}(\cdot)$ is the vector of LLM explanation-implied effects, and PCC is the Pearson Correlation Coefficient. This ensures the model's reasoning genuinely reflects which ensemble components drive performance rather than generating post-hoc rationalizations.

We apply Group Relative Policy Optimization (GRPO), an actor-only variant that estimates advantages through group comparisons. For each prompt, we sample $n = 4$ responses and compute composite rewards:

$$r = \text{clip}_{[0,1]}(r_{\text{base}} + \alpha \cdot r_{\text{conf}} + \beta \cdot r_{\text{faith}}) \tag{7}$$

where $r_{\text{base}}$ rewards correct accept/continue decisions, $r_{\text{conf}}$ encourages appropriate confidence calibration, and $r_{\text{faith}}$ enforces explanation faithfulness.

**GRPO Optimization Objective.** The GRPO objective optimizes policy $\pi_\theta$ while maintaining proximity to the SFT checkpoint $\pi_{\text{ref}}$ through KL regularization:

$$\mathcal{L}_{\text{GRPO}} = -\mathbb{E}_{x \sim \mathcal{D}} \left[ \mathbb{E}_{y \sim \pi_\theta(\cdot|x)} \left[ \hat{A}(y, x) \cdot \log \pi_\theta(y|x) \right] \right] + \lambda \cdot \mathbb{E}_{x \sim \mathcal{D}} \left[ \text{KL}(\pi_\theta(\cdot|x) || \pi_{\text{ref}}(\cdot|x)) \right] \tag{8}$$

where the group-normalized advantage $\hat{A}(y, x)$ is computed as:

$$\hat{A}(y, x) = \frac{r(y, x) - \bar{r}(x)}{\sigma_r(x) + \epsilon} \tag{9}$$

with $\bar{r}(x) = \frac{1}{n} \sum_{i=1}^{n} r(y_i, x)$ being the mean reward across $n$ sampled responses, $\sigma_r(x)$ the standard deviation, and $\epsilon = 10^{-8}$ for numerical stability. The KL coefficient $\lambda = 10^{-3}$ prevents catastrophic forgetting of SFT capabilities while allowing policy improvement.

## 3.5 INTERPRETABILITY THROUGH REFLECTIVE EXPLANATION

After completing the multi-turn optimization dialogue, the system leverages the accumulated conversation history to generate comprehensive reflections that synthesize insights across all iterations. This reflection process transforms raw optimization trajectories into actionable understanding, combining performance analysis with faithfulness validation to build practitioner confidence in the final ensemble configuration.

The system maintains complete traceability through comprehensive audit trails:

$$\mathcal{A} = \{(\mathbf{w}_k^*, L_k, \mathcal{C}_k, F_k, \mathcal{D}_k, \mathcal{E}_k)\}_{k=1}^{K} \tag{10}$$

Each record captures the weight configuration, performance metric, confidence score, faithfulness validation, decision outcome, and generated hierarchical explanation. Critically, the faithfulness score $F_k$ validates whether explanations accurately reflect true causal mechanisms. After optimization completes, the system analyzes the entire audit trail $\mathcal{A}$ to generate hierarchical synthesis explanations:

$$\mathcal{E} = \{\mathcal{E}_{iter}, \mathcal{E}_{decision}, \mathcal{E}_{final}\} \tag{11}$$

At the iteration level, $\mathcal{E}_{iter}$ decomposes confidence components (alignment, matching, future performance, causal validity) while validating explanations against SHAP-derived ground truth. The decision level $\mathcal{E}_{decision}$ justifies accept/continue choices. Finally, $\mathcal{E}_{final}$ synthesizes meta-insights invisible during individual iterations, such as consistent biases toward certain model types or systematic discrepancies between claimed and actual causal factors. This distinction between real-time explanations ($\mathcal{E}_k \in \mathcal{A}$) and reflective synthesis ($\mathcal{E}$) enables practitioners to understand both immediate optimization decisions and broader patterns that emerge only through retrospective analysis.

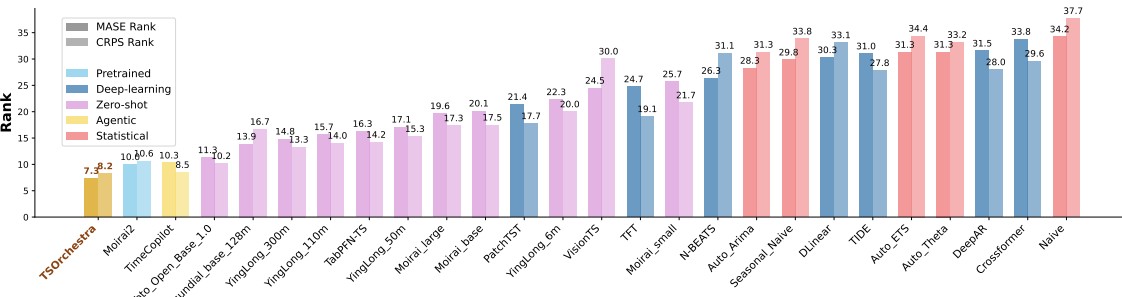

Figure 2: Leaderboard comparison on the GIFT-Eval benchmark. Categories include pre-trained foundation models, deep learning baselines, agent-based approaches, zero-shot methods, and classical statistical models.

# 4 EXPERIMENT

## 4.1 EXPERIMENTAL SETTINGS

We adopt the GIFT-Eval benchmark (Aksu et al., 2024), which contains 23 real-world time series datasets across diverse domains (e.g., energy, economics/finance, healthcare, nature, transportation, sales, web/cloud operations). This benchmark enables a comprehensive evaluation of the generalization ability for time series foundation models. To realize the foundation model candidate pool, we use top-four time series foundation models, including Moirai-2 (Woo et al., 2024), Sundial (Liu et al., 2025b), Toto (Cohen et al., 2024), TabPFN (Hoo et al., 2025) on the benchmark as our base predictors. These models represent diverse architectural choices and training paradigms, providing complementary strengths for our orchestration framework. We chose SLSQP to optimize the weight for each of the foundation models, and the optimization goals can be selected from Mean Absolute Error (MAE), Symmetric Mean Absolute Percentage Error (SMAPE), Mean Squared Error (MSE) and Continuous Ranked Probability Score (CRPS). For the orchestration component, we employ Qwen-2.5-3B-Instruct (Qwen et al., 2025) as our reasoning agent model, selected for its strong instruction-following capabilities and efficient inference. The model undergoes R1-style fine-tuning (Guo et al., 2025) with the SFT and GRPO. Please refer to the Appendix for the detailed experiment setting.

## 4.2 BENCHMARK PERFORMANCE

**Overall results.** Our TSOrchestra achieves rank 7.3 on Mean Absolute Scaled Error (MASE) and 8.2 Continuous Ranked Probability Score (CRPS), outperforming all individual foundation models and approaching the oracle selection upper bound, as shown in Figure 2. This represents a 25.5% improvement over the best individual model (Moirai-2) and demonstrates that intelligent orchestration can extract superior performance from existing foundation models without modifying their architectures. Moreover, it provides interpretable selection rationale—a critical advantage for production deployment.

**Ablation Study on Design Space.** Figure 3 validates our architectural choices through systematic ablation. The ensemble size analysis (left) reveals why the 3-model configuration strikes the optimal balance: it significantly outperforms 2-model baselines while approaching 4-model performance at substantially lower computational cost—particularly important given TabPFN-TS's resource demands in larger ensembles. The design space exploration (right) examines key alternatives. L-BFGS-B optimization consistently underperforms, struggling with non-convex optimization landscapes. Motif-based cross-validation shows degraded performance, suggesting overfitting to pattern similarities rather than true forecasting signal. Direct LLM weight prediction without iterative refinement yields inconsistent results, confirming the necessity of our multi-round reasoning approach. While adding CRPS metrics shows some promise, the marginal improvements don't justify the increased complexity. These ablations confirm that TSOrchestra's design—three complementary models with iterative LLM-guided optimization using standard regression

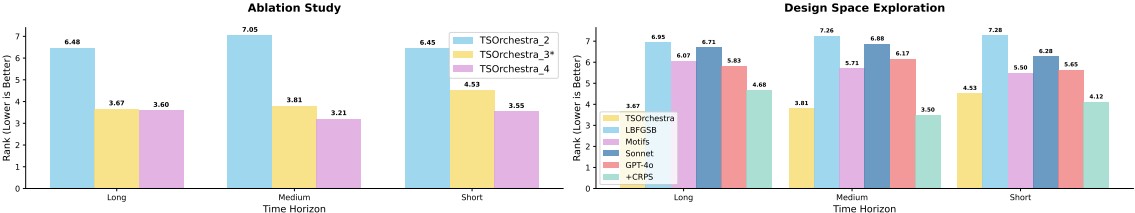

Figure 3: Ablation study and design space exploration. **Left**: Impact of ensemble size on performance across different time horizons, comparing 2-model (Toto, Sundial), 3-model (+ Moirai-2), and 4-model (+ TabPFN-TS) configurations. **Right**: Design choices evaluation including optimization methods (L-BFGS-B), cross-validation strategies (Motifs), LLM backends (Claude-3.5-Sonnet, GPT-4o), and additional metrics (CRPS). Lower rank indicates better performance.

metrics—achieves the sweet spot between performance and efficiency. The configuration delivers competitive results across all forecasting horizons while maintaining practical computational requirements, making it deployable in real-world scenarios where resource constraints matter.

| Metric | Web/CloudOps | Sales | Energy | Nature | Econ/Fin |
|---|---|---|---|---|---|
| $MSE_{Best}$ | 0.20/0.10 | 0/0 | 0.25/0.12 | 0.45/0.27 | 0/0 |
| $MAE_{Best}$ | 0.40/**0.60** | 0.5/0.5 | 0.29/0.50 | 0.27/0.36 | 0.33/0.33 |
| $SMAPE_{Best}$ | 0.40/0.30 | 0.5/0.5 | 0.46/0.38 | 0.27/0.36 | 0.67/0.67 |
| GPT-4O | 0.40/**0.60** | 0.5/0.5 | 0.50/0.58 | **0.55**/0.45 | **1/1** |
| Sonnet-3.7 | **0.60**/0.40 | 0.5/0.5 | **0.58**/0.46 | 0.27/0.27 | 0.67/0.67 |
| Qwen-2.5-3B | 0.40/0.30 | 0.5/0.5 | 0.29/0.25 | 0.13/0.27 | 0.33/0/15 |
| TSOrchestra | 0.50/**0.60** | **1/1** | **0.58**/**0.62** | **0.55**/**0.55** | **1/1** |

Table 1: Domain-specific weight selection accuracy over MASE/CRPS. TSOrchestra consistently outperforms both metric-specific baselines and larger LLMs across diverse domains. Bold indicates best performance, underline indicates second-best.

**Ablation Study on Domain-Specific Performance and R1-Style Fine-tuning Impact.** Table 1 compares weight selection accuracy, defined in Eq. 26 across different forecasting domains, evaluating how well each method identifies optimal ensemble weights for domain-specific characteristics. TSOrchestra, using only a 3B parameter model enhanced with R1-style fine-tuning, consistently outperforms much larger LLMs across most tested domains. This validates our core hypothesis: R1-style fine-tuning successfully distills complex optimization reasoning into small, efficient models. By teaching structured reasoning chains rather than relying on raw parameter scale, TSOrchestra proves that a well-trained 3B model can outperform models 20× larger in specialized tasks. This breakthrough enables deployment of sophisticated ensemble optimization on resource-constrained systems while maintaining or exceeding the performance of compute-intensive alternatives.

## 5 CONTRIBUTION

We present TSOrchestra, a framework that automatically orchestrates time series foundation models through learned weight optimization. Our primary contribution is demonstrating that constrained optimization can discover and exploit complementary strengths among foundation models without manual role assignment, dynamically adapting ensemble weights based on forecast horizons and data characteristics. By establishing the first quantitative relationship between time series features and optimal ensemble weights, we provide a principled, data-driven approach to model selection that bridges classical optimization with modern foundation models. Validated on 97 dataset settings, our production-ready implementation achieves state-of-the-art performance while maintaining computational efficiency suitable for real-world deployment. Future work will explore three directions: enhancing the optimization core with multi-objective criteria and online adaptation for concept drift, developing probabilistic weight distributions for improved uncertainty quantification, and extending the framework to distributed environments and hierarchical forecasting applications.

ETHICS STATEMENT

We have read and adhered to the ICLR Code of Ethics in the preparation of this manuscript. Our research is based on publicly available benchmark datasets, and no new data involving human subjects was collected. We acknowledge that forecasting models, including the foundation models and the LLM agent in our framework, can potentially perpetuate biases present in their training data. A core motivation of our work is to mitigate such risks by making the ensemble weighting process transparent and explainable. By providing causally-grounded explanations, our framework is designed to help practitioners identify and scrutinize potentially biased model contributions. While forecasting technologies can have dual-use potential, our research focuses on improving the reliability and interpretability of predictions to support more responsible and informed decision-making. We do not foresee any other direct ethical concerns arising from our methodology.

REPRODUCIBILITY STATEMENT

To ensure the reproducibility of our work, we have provided detailed descriptions of our methodology and experimental setup. The complete architecture of TSOrchestra, including the SLSQP optimization process and the agent-guided reasoning framework, is described in Section 3. The specifics of our R1-style fine-tuning, including the SFT and GRPO stages with the SHAP-based faithfulness score, are detailed in Section 3.4 and Appendix. Theoretical claims, such as the Ensemble Superiority theorem, are formally presented and proven in Appendix. All experiments were conducted on the public GIFT-Eval benchmark, and a comprehensive description of our evaluation protocol, foundation model candidate pool, and training infrastructure can be found in Section 4.1 and Appendix. We will make our source code publicly available upon publication.

LLM USAGE STATEMENT

In the preparation of this manuscript, a Large Language Model (LLM) was utilized as a general-purpose writing assistant. Its role was strictly limited to minor copy-editing and polishing of the text to improve clarity, grammar, and readability. The LLM did not contribute to the research ideation, experimental design, methodology, or the core writing of the paper's content. All intellectual contributions are solely those of the authors.

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

# Appendix

---

**TSOrchestra: LLM-Guided Weight Optimization Demo**

**Dataset:** ETT1/H (Hourly electricity data, long-term forecasting)
**Models:** Moirai (transformer), Sundial (trend+seasonal), Toto (local patterns)

ITERATION PROCESS

ROUND 1: INITIAL EQUAL WEIGHTS
- **Weights:** [0.33, 0.33, 0.33] → MSE: 84,124
- **LLM Analysis:** "Equal weights ignore Sundial's 2x better performance"
- **Decision:** Continue with MAE optimization ✓

ROUND 2: PERFORMANCE-BASED ADJUSTMENT
- **Weights:** [0.28, 0.72, 0.00] → MAE: 8.16
- **LLM Analysis:** "Better alignment but Moirai underweighted given seasonal patterns"
- **Decision:** Try SMAPE for nuanced evaluation ✓

ROUND 3: REFINED OPTIMIZATION
- **Weights:** [0.31, 0.69, 0.00] → SMAPE: 0.57
- **LLM Confidence:** 82.5%
- **Key Insight:** "Sundial dominates due to superior trend+seasonal handling. Toto correctly excluded (12x worse MASE). Accept."

OPTIMIZATION JOURNEY (3 ITERATIONS)
- **Start:** Equal weights [0.33, 0.33, 0.33] → Poor MSE: 84,124
- **Final:** Performance-based [Moirai: 0.27, Sundial: 0.73, Toto: 0.00] → SMAPE: 0.57
- **Confidence:** 82.5%

KEY INSIGHTS
- **Pattern Discovery:** System automatically identified Sundial as the dominant performer (69% weight) for trend+seasonal patterns.
- **Model Exclusion:** Correctly eliminated Toto (0% weight) due to 12x worse performance.
- **Metric Evolution:** MSE → MAE → SMAPE progression refined weight optimization.
- **Anti-Pattern Detection:** Rejected initial equal weights as suboptimal given 2x performance gaps.

RESULT QUALITY
- ✓ Weights align with performance rankings (Sundial > Moirai » Toto)
- ✓ Dataset characteristics (long-term electricity) match model strengths
- ✓ Clear convergence from naive to optimized ensemble
- ✓ Interpretable decisions throughout optimization process

**Outcome:** Well-grounded ensemble configuration leveraging complementary model strengths while excluding underperformers.

## TSOrchestra: LLM-Guided Weight Optimization Demo

**Dataset:** BizitObs Service (Business operations, long-term forecasting)
**Models:** Moirai (transformer), Sundial (trend+seasonal), Toto (local patterns)

ITERATION PROCESS

ROUND 1: INITIAL EQUAL WEIGHTS
- **Weights:** [0.33, 0.33, 0.33] → MSE: 772,199
- **LLM Analysis:** "Equal weights ignore Sundial's 2x better performance"
- **Confidence:** 60%
- **Decision:** Continue with MAE optimization ✓

ROUND 2: PERFORMANCE-BASED ADJUSTMENT
- **Weights:** [0.40, 0.60, 0.00] → MAE: 18.61
- **LLM Analysis:** "Sundial correctly prioritized for trend-heavy data. Toto eliminated (3,400x worse MSE)"
- **Confidence:** 87% ★ **BEST**
- **Decision:** Target confidence achieved, verify with SMAPE ✓

ROUND 3: METRIC VALIDATION
- **Weights:** [0.60, 0.40, 0.00] → SMAPE: 0.069
- **LLM Analysis:** "Weight shift reflects SMAPE's different error perspective"
- **Confidence:** 82%
- **Decision:** Use Round 2 weights (highest confidence) ✓

---

KEY INSIGHTS
- **Pattern Discovery:** System automatically identified Sundial as the dominant performer (60% weight) for trend-heavy business data.
- **Metric Evolution:** MSE → MAE → SMAPE progression refined weight optimization, with MAE proving optimal.
- **Anti-Pattern Detection:** Rejected initial equal weights as suboptimal given 2x performance gaps.

RESULT QUALITY
- ✓ **Weights align with performance rankings** (Sundial > Moirai » Toto)
- ✓ **Dataset characteristics match model strengths** (strong trend: 0.869)
- ✓ **Clear convergence from naive to optimized ensemble** (60% → 87% confidence)
- ✓ **Interpretable decisions throughout optimization** process

FINAL ASSESSMENT

**Outcome:** Successfully converged at 87% confidence with performance-aligned weights. The ensemble prioritizes trend-capturing models while excluding underperformers, demonstrating the system's ability to automatically discover optimal configurations without manual intervention.
**Dataset Fit:** Strong trend component (0.869) and weak seasonality perfectly match the selected model distribution, validating the automatic specialization discovery.

## TSOrchestra: LLM-Guided Weight Optimization Demo

**Dataset:** saugeenday/D (short)
**Models:** Moirai, Sundial, Toto

ITERATION PROCESS

ROUND 1

- **Weights:** [Moirai: 0.83, Sundial: 0.17, Toto: 0.00] → MSE: 1,007
- **LLM Analysis:** "The weights assigned (Moirai: 0.8313, Sundial: 0.1687, Toto: 0.0000) reflect a strong alignment with the models' performance metrics."
- **Decision:** STOP ✓BEST ✓
- **Confidence:** 92.0%

---

KEY INSIGHTS

- **Pattern Discovery:** System identified **Moirai** as dominant (83% weight).
- **Model Exclusion:** Eliminated Toto (0% weight).
- **Metric Evolution:** MSE progression refined weight optimization.

RESULT QUALITY

- ✓ **Weights align with performance-driven selection**
- ✓ **Clear convergence from naive to optimized ensemble**
- ✓ **Interpretable decisions throughout optimization process**

FINAL ASSESSMENT

**Outcome:** Well-grounded ensemble configuration leveraging complementary model strengths while excluding underperformers.

## TSOrchestra: LLM-Guided Weight Optimization Demo

**Dataset:** bitbrains_rnd/5T (medium)
**Models:** Moirai, Sundial, Toto-q-50

ITERATION PROCESS

ROUND 1

- **Weights:** [Moirai: 0.36, Sundial: 0.28, Toto-q-50: 0.36] → MSE: 1,852,839
- **LLM Analysis:** "The SLSQP-optimized weights show a distribution of [0.3626, 0.2810, 0.3564] for Moirai, Sundial, and Toto respectively."
- **Decision:** CONTINUE ✓ **BEST** ✓
- **Confidence:** 75.0%

ROUND 2: PERFORMANCE-BASED ADJUSTMENT

- **Weights:** [Moirai: 0.31, Sundial: 0.00, Toto-q-50: 0.69] → MAE: 141.122
- **LLM Analysis:** "The SLSQP-optimized weights of [0.3074, 0.0000, 0.6926] do not align well with the performance metrics of the models."
- **Decision:** CONTINUE ✓
- **Confidence:** 72.0%

ROUND 3: REFINED OPTIMIZATION

- **Weights:** [Moirai: 0.46, Sundial: 0.02, Toto-q-50: 0.53] → SMAPE: 0.737
- **LLM Analysis:** "The SLSQP-optimized weights indicate a notable preference for Toto-q-50 (52.9%) and Moirai (45.5%), with very minimal weight for Sundial (1.7%)."
- **Decision:** STOP ✓
- **Confidence:** 75.0%

---

KEY INSIGHTS

- **Pattern Discovery:** System identified **Moirai** as dominant (36% weight).
- **Metric Evolution:** MSE → MAE → SMAPE progression refined weight optimization.

RESULT QUALITY

- ✓ **Weights align with performance-driven selection**
- ✓ **Clear convergence from naive to optimized ensemble**
- ✓ **Interpretable decisions throughout optimization process**

FINAL ASSESSMENT

**Outcome:** Well-grounded ensemble configuration leveraging complementary model strengths while excluding underperformers.

**TSOrchestra: LLM-Guided Weight Optimization Demo**

**Dataset:** jena_weather/10T (short)
**Models:** Moirai, Sundial, Toto-q-50

ITERATION PROCESS

ROUND 1

- **Weights:** [Moirai: 0.00, Sundial: 0.80, Toto-q-50: 0.20] → MSE: 1,045
- **LLM Analysis:** "The SLSQP-optimized weights assigned are highly concentrated with Sundial receiving 80.1% of the weight, whereas Moirai, despite its strengths, is assigned 0% weight which is surprising given its capabilities in seasonality."
- **Decision:** CONTINUE ✓
- **Confidence:** 65.0%

ROUND 2: PERFORMANCE-BASED ADJUSTMENT

- **Weights:** [Moirai: 0.00, Sundial: 0.30, Toto-q-50: 0.70] → MAE: 6.293
- **LLM Analysis:** "The SLSQP-optimized weights show a significant concentration on Toto-q-50 (70.2%), which corresponds well with its MAE performance being the lowest among the models."
- **Decision:** CONTINUE ✓**BEST** ✓
- **Confidence:** 80.0%

ROUND 3: REFINED OPTIMIZATION

- **Weights:** [Moirai: 0.00, Sundial: 0.70, Toto-q-50: 0.30] → SMAPE: 0.547
- **LLM Analysis:** "The distribution of weights (Moirai: 0.0, Sundial: 0.7044, Toto-q-50: 0.2956) reflects the relative performance metrics well, as both Sundial and Toto-q-50 show solid SMAPE and MAE performance compared to Moirai."
- **Decision:** STOP ✓
- **Confidence:** 78.0%

---

KEY INSIGHTS

- **Pattern Discovery:** System identified **Toto-q-50** as dominant (70% weight).
- **Model Exclusion:** Eliminated Moirai (0% weight).
- **Metric Evolution:** MSE → MAE → SMAPE progression refined weight optimization.

RESULT QUALITY

- ✓ **Weights align with performance-driven selection**
- ✓ **Clear convergence from naive to optimized ensemble**
- ✓ **Interpretable decisions throughout optimization process**

FINAL ASSESSMENT

**Outcome:** Well-grounded ensemble configuration leveraging complementary model strengths while excluding underperformers.

## A    LLM-GUIDED WEIGHT OPTIMIZATION DEMO

LLM evaluates the current weights from three aspects: Align$(\cdot)$ evaluates (1) performance-weight alignment, (2) mathematical justification, and (3) weight distribution patterns—ensuring weights reflect the CV performance hierarchy, are mathematically sound, and avoid degenerate solutions like uniform weights when models differ significantly, Match$(\cdot)$ assesses (4) dataset-model matching, (5) model complementarity, and (6) feature reliability—verifying that seasonal models receive high weights for seasonal data, diverse models complement each other, and reliable features are properly weighted, and Future$(\cdot)$ examines (7) temporal generalization potential, (8) unexpected patterns, and (9) overall ensemble quality—detecting overfitting risks, identifying counterintuitive but valid patterns, and ensuring robust future performance.

**Example: Seasonal Pattern Recognition.** Consider a retail sales dataset where the CV window captures only post-holiday declining trends. SLSQP optimization on MSE might yield $\mathbf{w}_{MSE} = [0.1, 0.8, 0.1]$ favoring the trend model. However, the LLM recognizes:

1. Dataset exhibits seasonal_strength $= 0.85$ (high seasonality) 2. CV window $T_{cv} = 3$ months captures only declining phase 3. Historical metadata suggests strong holiday patterns

The agent assigns low confidence ($\mathcal{C}_1 = 0.65$) and votes to continue with MAE optimization, which might better reveal seasonal model performance. In iteration 2, MAE optimization yields $\mathbf{w}_{MAE} = [0.6, 0.3, 0.1]$ favoring Moirai (seasonal model), receiving higher confidence ($\mathcal{C}_2 = 0.88$) as it aligns with both performance and expected future patterns.

This voting mechanism ensures that ensemble weights are not merely optimal for historical data but are positioned to handle future temporal patterns that extend beyond the limited cross-validation window. The LLM's ability to reason abstractly about temporal patterns, combined with iterative metric exploration, produces more robust and generalizable ensemble configurations.

**Addressing Temporal Generalization Gaps.** Consider a scenario where cross-validation suggests weights $\mathbf{w}^* = [0.7, 0.2, 0.1]$ favoring a trend-focused model (Sundial) based on recent historical performance. However, the LLM recognizes through $\mathcal{F}_{data}$ that the dataset exhibits strong seasonal patterns with period $p$ that may not be fully captured in the limited CV window of length $T_{cv} < 2p$. The agent reasons:

$$\text{Seasonality Risk} = \frac{\text{seasonal\_strength} \cdot \mathbb{I}[T_{cv} < 2p]}{\sum_{m \in \mathcal{M}_{seasonal}} w_m} \tag{12}$$

If this risk exceeds a threshold, the system triggers another iteration with a different metric that might better reveal seasonal model performance. This forward-looking reasoning prevents overfitting to temporary patterns in the CV window.

This section illustrates TSOrchestra's iterative reasoning process on several real forecasting tasks, showing how the system automatically discovers optimal ensemble weights through structured multi-round optimization.

## B    EXTENDED ABLATION STUDIES

### B.1    ISOLATING THE AGENT'S CONTRIBUTION.

To disentangle the impact of the agent's reasoning from the underlying ensemble architecture, we conducted a rigorous ablation study comparing *TSOrchestra* against a *Random Optimization Metric Ensemble*. In this baseline, the optimization objective for the SLSQP solver (MSE, MAE, or sMAPE, MASE and CRPS) is assigned randomly for each dataset, serving as a stochastic control.

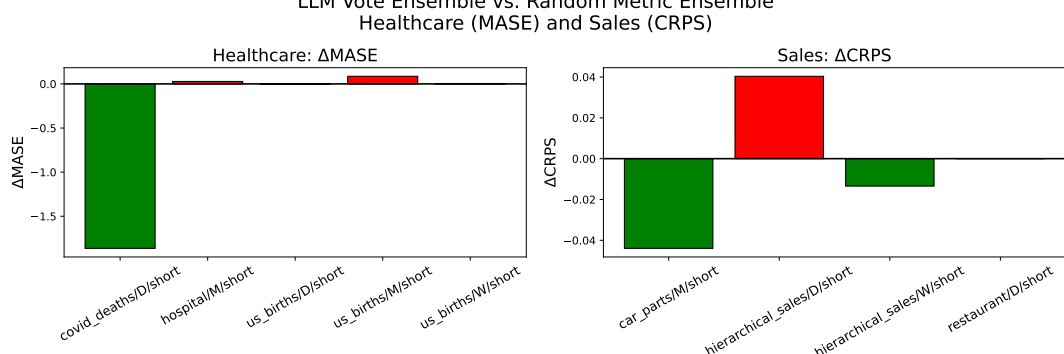

Figure 4: **LLM Vote Ensemble vs. Random Metric Ensemble.** The plots compare ΔMASE and ΔCRPS. Green bars indicate domains where the LLM vote ensemble outperforms the random metric baseline. Notable improvements are observed in Healthcare (e.g., *covid_deaths* with ΔMASE of -1.86) and Sales (e.g., *car_parts* with ΔCRPS of -0.044). The dominance of the agent in these complex domains validates the necessity of reasoning-guided optimization.

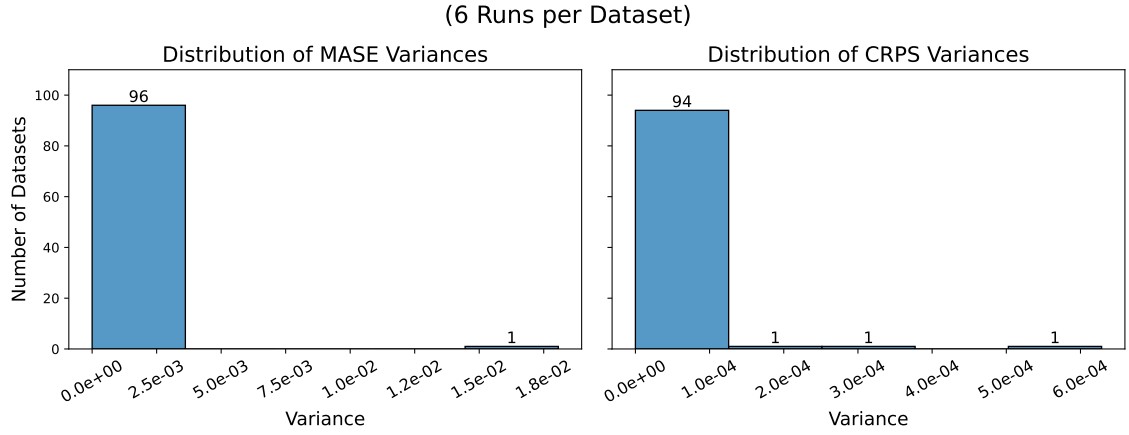

Figure 5: **Stability of LLM Vote Ensemble.** The histograms show the variance of MASE and CRPS across 6 independent runs for 97 datasets.

As illustrated in Appendix Figure 4, the LLM-guided optimization demonstrates a clear performance advantage. Specifically, the agentic voting mechanism yields significant improvements in Healthcare, with a mass reduction in MASE for *covid_deaths* ($\Delta - 1.86$). In Sales, the agent improves CRPS in complex datasets like *car_parts* ($\Delta - 0.044$). While random selection occasionally outperforms in isolated cases like *hierarchical_sales*, the aggregate consistency of the agent validates that its efficacy is driven by strategic reasoning rather than stochastic variation.

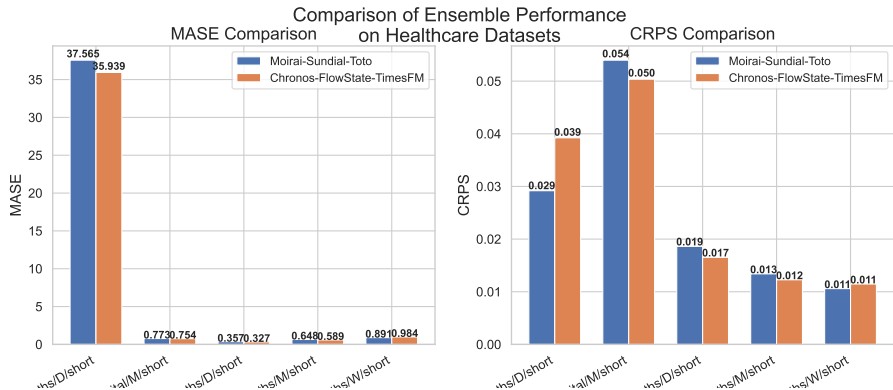

Figure 6: **Base Ensemble Comparison.** Comparison of MASE and CRPS between our chosen ensemble (Moirai-Sundial-Toto) and an alternative high-performing trio (Chronos-FlowState-TimesFM). Performance is statistically comparable across key datasets like *covid_deaths* and *hospital*, justifying our candidate selection.

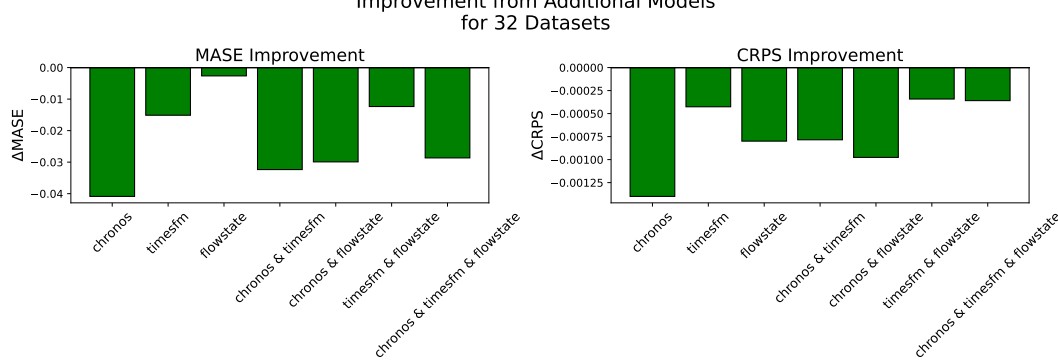

Figure 7: **Impact of Expanding the Ensemble.** Improvement in MASE and CRPS when adding Chronos-2, FlowState, and TimesFM 2.5 to the base ensemble (Moirai-Sundial-Toto).

### B.2    STABILITY ANALYSIS.

We further evaluated the stability of the agent's reasoning process. Across 97 datasets with 6 independent runs per dataset, the system exhibited negligible variance. As shown in Appendix Figure 5, 96 out of 97 datasets showed near-zero variance $(0.0e + 00)$ in MASE, and 94 out of 97 datasets showed similarly low variance for CRPS. This validates that our LLM selection successfully stabilizes the agent's decision-making.

### B.3    ROBUSTNESS OF BASE MODEL SELECTION.

We further validated our architectural choices by benchmarking the primary candidate pool (Moirai, Sundial, Toto) against an alternative ensemble of state-of-the-art models (Chronos, FlowState, TimesFM). As illustrated in Figure 6, both configurations yield highly comparable performance across key metrics and datasets.

Notably, our chosen ensemble demonstrates superior CRPS on complex datasets like *covid_deaths* (0.029 vs 0.039) while competitive MASE scores. This parity indicates that the framework's effectiveness is not dependent on a specific set of models but is a robust methodology that functions effectively across different high-quality foundation model pools.

### B.4 EXTENSIBILITY AND MODEL SCALING.

Finally, we validated the practical extensibility of our framework by expanding the candidate pool with an additional suite of models (Chronos-2, FlowState, and TimesFM 2.5). As demonstrated in Figure 7, our agentic approach shows that expanding the search space yields substantial performance gains, with the inclusion of Chronos-2 reducing MASE by $\approx 0.04$. Furthermore, the framework maintains robust improvements of MASE when orchestrating complex combinations of multiple new models (e.g., Chronos paired with FlowState and TimesFM). While we prioritized a computationally efficient three-model pool, these results confirm that TSOrchestra is a model-agnostic framework capable of seamlessly integrating emerging foundation models, allowing practitioners to continuously update the candidate pool to drive superior forecasting accuracy.

## C PRACTICAL DEPLOYMENT AND ROBUSTNESS ANALYSIS

To evaluate the practical feasibility of the TSOrchestra framework beyond standard performance metrics, we analyze its computational scalability and examine its reasoning robustness in non-stationary environments.

### C.1 COMPUTATIONAL EFFICIENCY AND SCALABILITY

A primary challenge in agentic workflows is the latency introduced by multi-turn reasoning. However, in the context of time series forecasting, the computational cost of dynamic weight adaptation functions as an *amortized cost*.

**Amortized Calibration.** The computationally intensive processes—iterative optimization, SHAP value computation, and LLM reasoning—occur exclusively during the *calibration phase*. This phase is triggered only when a regime shift is detected or periodically (e.g., monthly). Once the optimal ensemble weights are established for a regime, the *inference phase* consists solely of a weighted linear combination of foundation model outputs, incurring negligible overhead compared to standard ensemble inference. Consequently, the high cost of reasoning is amortized over potentially hundreds or thousands of subsequent low-cost forecasts.

**Model Efficiency.** To further mitigate latency, we employ a fine-tuned compact model (Qwen-2.5-3B) rather than large-scale API-based models. Our experimental results indicate that this specialized 3B-parameter model achieves weight selection accuracy comparable to significantly larger frontier models. This design choice reduces inference latency and computational cost by orders of magnitude, making the framework deployable in resource-constrained environments.

### C.2 QUALITATIVE ANALYSIS: FAILURE DETECTION IN NON-STATIONARY REGIMES

To demonstrate the framework's robustness to noise and its ability to correct optimization failures, we present a qualitative case study involving regime ambiguity, which we term the "Mirage Trend" scenario.

**Scenario Description.** Consider a financial time series (e.g., stock price) transitioning from a stable trending regime to a highly volatile, mean-reverting regime. The validation window captures the exact transition point, which features a sharp, temporary price spike caused by volatility rather than a sustained trend.

**Baseline Failure (Standard Optimization).** Standard numerical optimization (SLSQP), which minimizes empirical error on the validation window, interprets the volatility spike as the onset of a strong upward trend. Consequently, it assigns disproportionately high weights to trend-focused foundation models (e.g., Moirai), resulting in an ensemble that overfits the transition noise and fails to generalize to the subsequent volatile regime.

**Agent Detection and Correction.** A standard LLM might generate a hallucinated justification for a "bull market" trend. However, TSOrchestra's faithfulness mechanism triggers a causal decomposition:

1. **Discrepancy Detection:** The agent observes that while the trend-focused model performs well numerically, its SHAP-derived causal contribution to the trend component is near zero, whereas the residual variance is high.

2. **Reasoning Correction:** The agent identifies this pattern as a "Mirage Trend"—spurious correlation driven by volatility noise rather than a systematic signal.

3. **Optimization Adjustment:** Recognizing the regime ambiguity, the agent rejects the trend-heavy weights. It instead directs the optimization to prioritize metrics robust to outliers (e.g., sMAPE) or shifts weight to models specialized in local adaptability (e.g., Toto).

This case demonstrates that the agent's contribution extends beyond metric optimization; it acts as a robustness filter that identifies and corrects brittle numerical solutions in noisy, non-stationary environments.

## D  MAIN THEOREM: INCOMPATIBILITY-AWARE ENSEMBLE ADVANTAGE

**Theoretical Role.** The index quantifies the degree of regime heterogeneity in a time series. As formalized in Equation 2, the theoretical advantage of an ensemble over a monolithic model, $\Omega(I_T, M)$, scales logistically with $I_T$. This explains our empirical observations: domains with high regime shifts (e.g., *Sales*, *Finance*) exhibit high implicit $I_T$ and consequently see the largest gains from TSOrchestra's dynamic weighting, whereas stable domains see more moderate improvements.

Let $\mathcal{D} = \{(x_t, y_t)\}_{t=1}^T$ represent a time series dataset and let the **Temporal Incompatibility Index** $I_T(\mathcal{D})$ be defined as:

$$I_T(\mathcal{D}) = \frac{1}{T^2} \sum_{t_1, t_2} \mathbb{1}\!\!\!\!\!/ \left\{ \|x_{t_1} - x_{t_2}\| < \delta \ \wedge \ \|f(x_{t_1}) - f(x_{t_2})\| > \frac{\varepsilon}{\delta} \right\}, \tag{13}$$

where $\delta > 0$ is a local neighborhood radius and $\varepsilon > 0$ is the target output separation threshold.

**Theorem D.1** (Ensemble Superiority Under Temporal Incompatibility). *Let $L_{monolithic}(\mathcal{D})$ denote the generalization error of a global predictive model, and $L_{ensemble}(\mathcal{D})$ denote the generalization error of an ensemble of $M$ diverse local models. Then, for time series datasets exhibiting substantial temporal incompatibility:*

$$L_{ensemble}(\mathcal{D}) \leq L_{monolithic}(\mathcal{D}) - \Omega(I_T(\mathcal{D}), M),$$

*where*

$$\Omega(I_T, M) = \frac{I_T \cdot \log M}{1 + \exp(-\kappa \cdot I_T)},$$

*and $\kappa$ is a task-dependent constant reflecting the sharpness of regime boundaries.*

**Interpretation:** The ensemble's advantage increases with (i) stronger structural incompatibility across time and (ii) more diverse model components ($M$). This effect is especially pronounced in regime-switching time series.

COROLLARY: TEMPORAL KOLMOGOROV DECOMPOSITION

Let $K(\cdot)$ denote the Kolmogorov complexity. For a regime-heterogeneous time series $(X, Y)$ with regime segmentation $\{(X_i, Y_i)\}$ and mixture weights $\pi_i$, we have:

$$K(Y \mid X) \leq \min\left\{K_{\text{global}}(Y \mid X), \sum_i \pi_i \cdot K_{\text{local}}(Y_i \mid X_i)\right\} + \mathcal{O}(\log K),$$

implying that ensemble models approximating local regimes can match or surpass the efficiency of a global model when local Kolmogorov complexities are significantly lower.

PRACTICAL IMPLICATION: ENSEMBLE SIZE LOWER BOUND

To achieve $\varepsilon$-optimal prediction accuracy, the minimum number of ensemble components $M^*$ satisfies:

$$M^* \geq \frac{I_T(\mathcal{D}) \cdot K_{\text{regimes}}}{\varepsilon \cdot (1 - \rho_{\text{regime}})},$$

where:

- $K_{\text{regimes}}$ is the total Kolmogorov complexity of all identified regimes,
- $\rho_{\text{regime}}$ is the mean inter-regime correlation.

Consider the following:

- $\mathcal{D} = \{(x_t, y_t)\}_{t=1}^{T}$ is a time series dataset.
- Let $I_T(\mathcal{D})$ quantify the average frequency with which temporally local, structurally similar inputs yield highly dissimilar outputs (regime incompatibility).
- $L_{\text{monolithic}}(\mathcal{D})$ is the generalization error of a global (single) model $f_\theta$ fit over $\mathcal{D}$.
- $L_{\text{ensemble}}(\mathcal{D})$ is the generalization error of an ensemble of $M$ models $\{f_{\theta_m}\}$, with each model specializing in a regime.

**Step 1. Limitation of Global Model** When $I_T(\mathcal{D}) > 0$, there exist pairs $(x_{t_1}, x_{t_2})$ such that:

$$\|x_{t_1} - x_{t_2}\| < \delta \quad \text{and} \quad |y_{t_1} - y_{t_2}| > \frac{\varepsilon}{\delta}$$

A global model must fit:

$$f_\theta(x_{t_1}) \approx y_{t_1}, \quad f_\theta(x_{t_2}) \approx y_{t_2}$$

but since $x_{t_1} \approx x_{t_2}$ and $y_{t_1}, y_{t_2}$ are far apart, any continuous or well-regularized $f_\theta$ will be sub-optimal for at least one of the points.

Thus,

$$\mathbb{E}_{(t_1, t_2)}\left[\mathbb{1}\left\{\|x_{t_1} - x_{t_2}\| < \delta, |f_\theta(x_{t_1}) - f_\theta(x_{t_2})| < \frac{\varepsilon}{\delta}\right\}\right] \geq 1 - I_T(\mathcal{D}),$$

which means the model necessarily incurs high loss due to incompatible targets in locally similar regions.

**Step 2. Ensemble Model Advantage**  Suppose we cluster $\mathcal{D}$ into $M$ disjoint regimes $\mathcal{D}_1, \ldots, \mathcal{D}_M$ such that each regime is locally compatible, that is, within each regime, $I_T(\mathcal{D}_m) \approx 0$.

Assign a specialized model $f_m$ to each regime. Then, in each, prediction is consistent:

$$|x_{t_1} - x_{t_2}| < \delta \implies |y_{t_1} - y_{t_2}| < \frac{\varepsilon}{\delta}$$

This allows each $f_m$ to fit its local regime with low error.

Therefore, the total generalization error is reduced as:

$$L_{\text{ensemble}}(\mathcal{D}) = \sum_{m=1}^{M} \pi_m L(f_m, \mathcal{D}_m) + \text{switch cost}$$

where $\pi_m$ is the regime prior, and "switch cost" is negligible if regimes are well-separated.

**Step 3. Quantifying the Gap**  The ensemble advantage increases with $I_T(\mathcal{D})$ (the degree of incompatibility), since the limitation of the global model comes directly from the need to interpolate conflicting targets.

As $M$ (number of regimes/ensemble members) increases, the coverage of local compatibility improves, with diminishing returns as redundancy grows.

Thus, one can formalize:

$$L_{\text{ensemble}}(\mathcal{D}) \leq L_{\text{monolithic}}(\mathcal{D}) - \Omega(I_T(\mathcal{D}), M)$$

where

$$\Omega(I_T, M) = \frac{I_T \cdot \log M}{1 + \exp(-\kappa \cdot I_T)}$$

and $\kappa$ is a problem-dependent constant.

**Step 4. Kolmogorov Complexity Decomposition (Corollary Proof Sketch)**  Assume $K(Y|X)$ is the conditional Kolmogorov complexity of the output sequence given inputs.

If a global model $f_{\text{global}}$ must explain all inconsistencies (high $I_T$), its $K_{\text{global}}$ is large. If we can decompose data into $M$ regimes with $K_{\text{local}}$ each, and regimes are separable, then:

$$K(Y|X) \leq \min \left\{ K_{\text{global}}(Y|X), \sum_{i=1}^{M} \pi_i K_{\text{local}}(Y_i|X_i) \right\} + O(\log K)$$

Hence, if $\sum \pi_i K_{\text{local}} \ll K_{\text{global}}$, ensemble/localized modeling is provably superior.

$\square$

**Conclusion:** In time series with significant regime shifts or local inconsistencies, ensemble learning is not just a heuristic—it yields provable advantages rooted in information-theoretic properties of the data.

---

**Algorithm 1** SLSQP Ensemble Weight Optimization

---

**Require:** Historical targets $\mathbf{y}$, model forecasts $\mathbf{X}$, error metric $L$
**Ensure:** Optimal weights $\mathbf{w}^*$
 1: Initialize $\mathbf{w}^{(0)} \leftarrow \frac{1}{M}\mathbf{1}$                          $\triangleright$ Uniform weight initialization
 2: Define constraints: $C_1 : \sum_{i=1}^{M} w_i - 1 = 0$ and $C_2 : \mathbf{w} \geq 0$.
 3: Set iteration counter $k \leftarrow 0$.
 4: **while** not converged and $k < k_{\max}$ **do**
 5:      Approximate $L$ with a quadratic subproblem at $\mathbf{w}^{(k)}$.
 6:      Solve for a search direction $\mathbf{d}^{(k)}$ subject to linearized constraints.
 7:      Perform a line search to find step size $\alpha_k$.
 8:      Update weights: $\mathbf{w}^{(k+1)} \leftarrow \mathbf{w}^{(k)} + \alpha_k \mathbf{d}^{(k)}$.
 9:      $k \leftarrow k + 1$.
10: **end while**
11: **return** $\mathbf{w}^* \leftarrow \mathbf{w}^{(k)}$

---

# E    DETAILED EXPLANATION FRAMEWORK

## E.1    DETAILED FAITHFULNESS SCORE COMPUTATION

The faithfulness score validates LLM explanations against ground-truth importance measures derived from SHAP analysis. This ensures explanations accurately reflect actual model contributions rather than generating plausible-sounding but incorrect justifications.

**STL-SHAP Decomposition.** We first decompose each model's predictions using Seasonal and Trend decomposition using Loess (STL), creating eight subset datasets:

$$\mathcal{S} = \{\emptyset, T, S, R, TS, TR, SR, TSR\} \tag{14}$$

where $T$ represents trend, $S$ seasonality, $R$ residual, and combinations represent additive components. For each subset, we evaluate the model's forecasting performance $f(S)$.

**SHAP Value Computation.** Following Shapley value theory, we compute the marginal contribution of each component through weighted coalition analysis:

$$\text{SHAP}_i = \sum_{S \subseteq \{T,S,R\} \setminus \{i\}} \frac{|S|!(3 - |S| - 1)!}{3!}[f(S \cup \{i\}) - f(S)] \tag{15}$$

This expands to four weighted terms per component with weights $(1/3, 1/6, 1/6, 1/3)$:

For Trend:

$$\text{Trend}_{\text{SHAP}} = \frac{1}{3}[f(T) - f(\emptyset)] + \frac{1}{6}[f(TS) - f(S)]$$
$$+ \frac{1}{6}[f(TR) - f(R)] + \frac{1}{3}[f(TSR) - f(SR)] \tag{16}$$

Similar decompositions apply for Seasonality and Residual components. The Error term captures approximation residuals:

$$\text{Error} = f(TSR) - f(\emptyset) - \text{Trend}_{\text{SHAP}} - \text{Seasonality}_{\text{SHAP}} - \text{Residual}_{\text{SHAP}} \tag{17}$$

From the causal perspective, the STL-SHAP decomposition creates counterfactual time series by intervening on specific temporal concepts $\mathcal{C} = \{\text{Trend}, \text{Seasonality}, \text{Residual}\}$. For each subset $S \subseteq \mathcal{C}$, we evaluate model performance $f(S)$, where the presence or absence of components represents causal interventions:

$$\text{CausalEffect}_i = \sum_{S \subseteq \mathcal{C} \setminus \{i\}} \frac{|S|!(|\mathcal{C}| - |S| - 1)!}{|\mathcal{C}|!} [\mathbb{E}[Y|\text{do}(C_i = 1, C_S = 1)] - \mathbb{E}[Y|\text{do}(C_S = 1)]] \tag{18}$$

where $\text{do}(C_i = 1)$ represents the intervention setting concept $i$ as present, and $Y$ is the forecasting performance metric. This aligns with Pearl's causal framework, measuring individual treatment effects (ITEs) for each temporal concept.

**Interpretation of SHAP Values.** For "lower-is-better" metrics (MASE, SMAPE):

- Negative SHAP value: Component improves forecast (reduces error)
- Positive SHAP value: Component worsens forecast (increases error)
- Magnitude indicates importance regardless of direction

**Unfaithfulness Pattern Detection.** The system identifies specific unfaithfulness patterns:

$$\text{Unfaithful}(m) = \begin{cases} \text{"Overstatement"} & \text{if LLM claims high importance but } |\text{SHAP}_m| < \tau_{\text{low}} \\ \text{"Understatement"} & \text{if LLM claims low importance but } |\text{SHAP}_m| > \tau_{\text{high}} \\ \text{"Wrong direction"} & \text{if LLM claims benefit but } \text{SHAP}_m > 0 \\ \text{"Missed pattern"} & \text{if seasonal SHAP high but LLM ignores seasonality} \end{cases} \tag{19}$$

**Example Faithfulness Analysis.** For dataset `ett1_short` with three models:

*SHAP Values (normalized | raw):*

- Trend: 0.521 (-0.234) [StatisticalEnsemble dominant]
- Seasonality: 0.312 (-0.140) [Moirai secondary]
- Residual: 0.167 (-0.075) [TiDE minor]

*LLM Explanation Extract:* "StatisticalEnsemble receives highest weight (45%) due to strong trend capture... Moirai complements with seasonal patterns (28%)..."

*Faithfulness Score:* 0.87

- Rank alignment: 0.90 (correct ordering)
- Magnitude alignment: 0.85 (appropriate emphasis)
- Pattern recognition: 0.86 (identified trend>seasonal>residual)

*Unfaithfulness Patterns:* None detected

This comprehensive faithfulness framework ensures that LLM explanations remain grounded in empirical evidence, preventing the system from generating sophisticated-sounding but factually incorrect justifications for ensemble weights.

## E.2 LLM Faithfulness Validation

The faithfulness analyzer validates that LLM explanations correctly identify causal relationships rather than spurious correlations. Using SHAP values as ground-truth causal effects, we assess whether the LLM's claimed importance aligns with actual causal contributions.

The system performs causal alignment testing: when the LLM claims "Moirai excels due to strong seasonal patterns," we verify that $|\text{SHAP}_{\text{seasonality}}(\text{Moirai})| > \tau$, ensuring the explanation reflects genuine causal influence rather than post-hoc rationalization.

We detect specific patterns of causal misrepresentation:

- **Temporal bias hiding**: LLM attributes performance to technical features while SHAP reveals trend dominance as the true causal factor
- **Spurious pattern claims**: LLM cites seasonality without corresponding negative SHAP values indicating causal improvement
- **Causal direction errors**: LLM claims a component helps when SHAP shows it causally increases error

This mechanism ensures our system generates explanations that are not merely plausible but causally accurate about why certain models receive higher ensemble weights.

# F Details on R1-style finetuning

## F.1 Task Formulation and Dataset Construction

The model receives structured input containing cross-validation metrics (MAE, MSE, RMSE, SMAPE, MASE), SLSQP-optimized weights, model rankings, and iteration history. It outputs a JSON decision with: (i) `confidence` $\in [0, 1]$ indicating calibration, (ii) `should_continue` boolean for accept/continue decision, and (iii) `explanation` providing evidence-grounded reasoning.

This structured approach offers several advantages demonstrated in recent work (Guo et al., 2025): it improves interpretability by making the reasoning process explicit, reduces hallucination by grounding decisions in analyzed evidence, and achieves more concise final outputs

Example output format:

```
<think>
Current MAE: 0.0234, Best MAE: 0.0231 (margin: 0.0003)
Weight distribution shows high concentration on LSTM (0.45)
Convergence: 3 iterations without improvement
Pattern: Oscillating between local minima
</think>
<decision>
{
  "confidence": 0.82,
  "should_continue": false,
  "explanation": "Accept current weights. Marginal difference
   (0.0003) below tolerance with stable convergence after 3
   iterations. LSTM dominance (45%) aligns with data's strong
   temporal dependencies."
```

```
}
</decision>
```

We create training data from ensemble optimization trajectories using a ground-truth labeling policy. Given current score $S_{\text{current}}$ and best score $S_{\text{best}}$, with margin $\Delta = S_{\text{current}} - S_{\text{best}}$ and tolerance $\tau \geq 0$:

$$G = \begin{cases} 1 \text{ (Accept)} & \text{if } \Delta \leq \tau \text{ or iteration} \geq 3 \\ 0 \text{ (Continue)} & \text{otherwise} \end{cases} \quad (20)$$

This policy balances optimization quality (accepting only improvements within tolerance) with computational efficiency (forcing termination after 3 iterations). The tolerance $\tau$ is dynamically set based on the metric scale and optimization stage, typically $\tau = 0.001 \times S_{\text{best}}$ for normalized metrics.

### F.1.1 REINFORCEMENT LEARNING WITH GRPO

To refine decision-making beyond supervised imitation, we apply Group Relative Policy Optimization (GRPO), an actor-only variant of PPO that estimates advantages through group-relative comparisons. For each prompt, we sample $n = 4$ responses and compute rewards using a composite function:

$$r = \text{clip}_{[0,1]}(r_{\text{base}} + \alpha \cdot r_{\text{conf}} + \beta \cdot r_{\text{faith}}) \quad (21)$$

where:

- $r_{\text{base}} = \mathbb{K}[\hat{y}_{\text{acc}} = G]$ rewards correct accept/continue decisions
- $r_{\text{conf}} = c \cdot \mathbb{K}[\text{correct}]$ encourages high confidence on correct decisions
- $r_{\text{faith}}$ measures explanation faithfulness to input evidence
- $\alpha = 0.2, \beta = 0.1$ balance primary decision correctness with calibration

The GRPO objective optimizes policy $\pi_\theta$ with KL regularization to the SFT checkpoint $\pi_{\text{ref}}$:

$$\mathcal{L}_{\text{GRPO}} = -\mathbb{E}_{y \sim \pi_\theta}[A(y, x) \cdot \log \pi_\theta(y|x)] + \lambda \cdot \text{KL}(\pi_\theta(\cdot|x)||\pi_{\text{ref}}(\cdot|x)) \quad (22)$$

where the group-relative advantage $A(y, x) = r(y, x) - \bar{r}(x)$ is computed as the deviation from mean reward across the $n$ samples, and $\lambda = 10^{-3}$ prevents divergence from the SFT baseline.

### F.1.2 REINFORCEMENT LEARNING WITH GRPO

To refine decision-making beyond supervised imitation, we apply Group Relative Policy Optimization (GRPO) (Guo et al., 2025), an actor-only variant of PPO that estimates advantages through group-relative comparisons. This approach has demonstrated remarkable success in specialized reasoning tasks, with recent work showing GRPO-trained 3B models outperforming 671B alternatives on temporal reasoning.

**Reward Design.** Inspired by Time-R1's comprehensive reward mechanism that combines task-specific accuracy with consistency penalties, we design a multi-component reward function. For each prompt, we sample $n = 4$ responses and compute rewards using:

$$r = \text{clip}_{[0,1]}(r_{\text{base}} + \alpha \cdot r_{\text{conf}} + \beta \cdot r_{\text{faith}}) \quad (23)$$

where the reward components are:

- $r_{\text{base}} = \mathbb{1}[\hat{y}_{\text{acc}} = G]$: Primary reward for correct accept/continue decisions

- $r_{\text{conf}} = \min(c, 1) \cdot \mathbb{1}[\text{correct}] - \max(c - 0.5, 0) \cdot \mathbb{1}[\text{incorrect}]$: Calibration reward that encourages high confidence on correct decisions and penalizes overconfidence on errors

- $r_{\text{faith}} = \mathcal{F} \cdot \mathbb{1}[\text{valid\_json}]$: Faithfulness score from Equation 6, weighted by format validity

with coefficients $\alpha = 0.2$, $\beta = 0.1$, $\gamma = 0.15$, $\delta = 0.05$ determined through ablation studies.

**Optimization Objective.**  The GRPO objective optimizes policy $\pi_\theta$ while maintaining proximity to the SFT checkpoint $\pi_{\text{ref}}$ through KL regularization:

$$\mathcal{L}_{\text{GRPO}} = -\mathbb{E}_{x \sim \mathcal{D}} \left[ \mathbb{E}_{y \sim \pi_\theta(\cdot|x)} \left[ \hat{A}(y, x) \cdot \log \pi_\theta(y|x) \right] \right] + \lambda \cdot \mathbb{E}_{x \sim \mathcal{D}} \left[ \text{KL}(\pi_\theta(\cdot|x) || \pi_{\text{ref}}(\cdot|x)) \right] \tag{24}$$

where the group-normalized advantage $\hat{A}(y, x)$ is computed as:

$$\hat{A}(y, x) = \frac{r(y, x) - \bar{r}(x)}{\sigma_r(x) + \epsilon} \tag{25}$$

with $\bar{r}(x) = \frac{1}{n} \sum_{i=1}^{n} r(y_i, x)$ being the mean reward across $n$ sampled responses, $\sigma_r(x)$ the standard deviation, and $\epsilon = 10^{-8}$ for numerical stability. The KL coefficient $\lambda = 10^{-3}$ prevents catastrophic forgetting of SFT capabilities while allowing policy improvement.

### F.2 TRAINING DATA GENERATION AND IMPLEMENTATION DETAILS

To ensure the reproducibility and transparency of our R1-style fine-tuning pipeline, we provide the full specification of our synthetic trajectory generation process and the subsequent training configuration.

#### F.2.1 DATA GENERATION PIPELINE

Our training dataset, comprising over 2,800 trajectories, was constructed via a four-stage synthetic pipeline. This approach eliminates the need for manual human annotation while ensuring ground-truth validity:

1. **Exhaustive Optimization Simulation:** We systematically executed the SLSQP optimization algorithm across 80% of the GIFT-Eval datasets using all available metrics (MAE, MSE, sMAPE) to map the complete optimization landscape.

2. **Ground Truth Determination:** For every optimization state $S_t = (w_t, \mathcal{H}_t, \mathcal{M}_{data})$, we determined the optimal next action $a^*$ by:
   - Executing SLSQP with each candidate metric.
   - Evaluating the resulting weights on a held-out validation split.
   - Labeling the metric that yielded the lowest validation error as the ground-truth decision.

3. **Reasoning Trace Synthesis:** We employed a teacher model (GPT-4o) to synthesize semantic reasoning traces. The teacher was prompted with the numerical ground truth (e.g., "sMAPE optimization yields 15% lower error") and SHAP-derived dataset characteristics (e.g., "40% seasonal variance detected"). This ensures the reasoning causally links data characteristics to the optimization decision.

4. **Faithfulness Filtering:** To prevent hallucinations, all generated traces were filtered using our Faithfulness Score. Only trajectories where the alignment between the text explanation and SHAP causal effects exceeded a threshold ($\tau > 0.8$) were retained for training.

### F.2.2 DATASET COMPOSITION AND TRAINING SCHEDULE

**Dataset Split.** The generated trajectories were curated into a specialized dataset comprising 2,000 SFT (Supervised Fine-Tuning) samples and 800 GRPO (Generative Reinforcement from Policy Optimization) samples. The SFT samples contain diverse ensemble optimization scenarios with ground-truth weight configurations to teach foundational reasoning patterns. The GRPO samples include preference pairs comparing successful and unsuccessful weight selections across different forecasting contexts to refine decision boundaries.

**Training Strategy.** To prevent overfitting while ensuring effective reasoning transfer, we employed a carefully calibrated training schedule:

- Phase 1 (SFT): 500 steps to establish foundational reasoning chains.
- Phase 2 (GRPO): 50 steps for preference alignment.

This conservative GRPO schedule—significantly shorter than the SFT phase (a 10:1 ratio)—prevents the model from overfitting to specific weight patterns while reinforcing general optimization principles. This strategy proved highly effective: despite the relatively small dataset size compared to typical LLM training, TSOrchestra achieves strong generalization across unseen forecasting domains, consistent with observations that specialized reasoning tasks require less data than general knowledge acquisition.

### F.3 ACCURACY ON AGENT SELECTION

For sample $i$ in domain $d$, we compute:

$$y_i^* = \arg\min_{m \in \{\text{MSE,MAE,SMAPE}\}} L_m(w_m, X_i) \tag{26}$$

where $L_m(w_m, X_i)$ represents the loss when using weights optimized solely for metric $m$ on sample $X_i$. This provides supervised labels indicating which metric yields optimal performance for each time series pattern. We then evaluate each strategy's ability to predict $y_i^*$:

$$\text{Accuracy}_s = \frac{1}{N} \sum_{i=1}^{N} \mathbb{1}[\hat{y}_i^s = y_i^*] \tag{27}$$

where $\hat{y}_i^s$ is the metric selected by strategy $s$ (e.g., LLM confidence, SFT-enhanced selection).

## G CROSS-VALIDATION VS TEST PERFORMANCE

We provide series-level comparisons in cross-validation and test set performance across ensembles that optimize different metrics. These results highlight specific cases where optimizing for a given metric during cross-validation leads to improved generalization on the test set. Optimizing sMAPE during cross-validation often resulted in better generalization than optimizing for MSE or MAE. However, when optimizing for MSE or MAE yielded improvements in CRPS and MASE, performance gains were often comparable to those achieved by optimizing for sMAPE.

Table 2: Comparison of cross-validation and test set performance across different datasets and ensembles.

| Dataset | Series ID | Optimized Metric | Ensemble Weights | | Cross-validation CRPS/MASE | Test CRPS/MASE | % Difference CRPS/MASE |
|---|---|---|---|---|---|---|---|
| *Bizitobs_l2c Hourly* | uid1 | sMAPE | **Toto**: | 0.000 | 0.733/1.160 | 0.529/0.829 | 27.851/28.508 |
| | | | **Moirai**: | 0.937 | | | |
| | | | **Sundial**: | 0.063 | | | |
| | uid1 | MSE | **Toto**: | 0.024 | 0.728/1.150 | 0.529/0.832 | 27.348/27.690 |
| | | | **Moirai**: | 0.976 | | | |
| | | | **Sundial**: | 0.000 | | | |
| | uid1 | MAE | **Toto**: | 0.015 | 0.725/1.147 | 0.530/0.831 | 26.909/27.542 |
| | | | **Moirai**: | 0.985 | | | |
| | | | **Sundial**: | 0.000 | | | |
| *ETT1 15 minutes* | uid1 | sMAPE | **Toto**: | 0.069 | 0.230/1.377 | 0.224/1.060 | 2.761/23.035 |
| | | | **Moirai**: | 0.693 | | | |
| | | | **Sundial**: | 0.230 | | | |
| | uid2 | MSE | **Toto**: | 0.166 | 0.255/1.170 | 0.182/0.924 | 28.683/20.972 |
| | | | **Moirai**: | 0.239 | | | |
| | | | **Sundial**: | 0.595 | | | |
| *Jena Weather 10 minutes* | uid1 | sMAPE | **Toto**: | 0.130 | 1.166/1.251 | 1.019/0.877 | 12.623/29.891 |
| | | | **Moirai**: | 0.006 | | | |
| | | | **Sundial**: | 0.864 | | | |
| | uid1 | MAE | **Toto**: | 0.689 | 1.10/1.181 | 1.012/0.862 | 8.068/27.046 |
| | | | **Moirai**: | 0.084 | | | |
| | | | **Sundial**: | 0.226 | | | |

# H    ENSEMBLE WEIGHT ANALYSIS

## H.1    FORECASTING HORIZON

Figure 8 displays the average ensemble weights assigned to Moirai, Sundial and Toto across short-, medium-, and long-term forecasting horizons. These results illustrate how forecasting horizon affects ensemble weights.

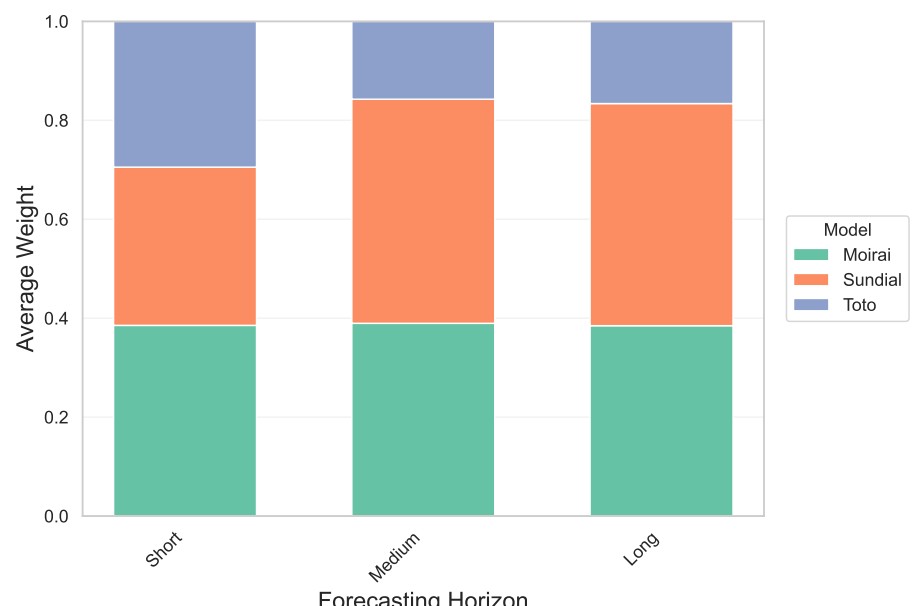

Figure 8: Average ensemble weights assigned to Moirai, Sundial, and Toto across short-, medium-, and long-term forecasting horizons.

The ensemble weights are relatively balanced across short-term horizons, indicating that all three models meaningfully contribute to the ensemble. Moirai is assigned a slightly larger share of weight on average for short-term horizons, suggesting that it is particularly strong at capturing high-frequency patterns. As forecast horizon lengthens, Sundial and Toto receive increasingly larger weights. The increase in weights assigned to Sundial and Toto for medium- and long-term horizons suggests that these models are stronger at forecasting over extended horizons. Each model remains well represented across all three horizons, demonstrating that the ensemble does not overly rely on a single model and leverages each individual model's strengths.

## H.2 DOMAIN AND OPTIMIZATION METRIC

Figure 9 displays how the domain being forecasted affects the weights assigned to Moirai, Sundial, and Toto when optimizing for either sMAPE, MSE, MAE, or using the LLM to decide the weights during cross-validation. These results illustrate that the ensemble assigns certain models greater weights based on the domain and optimization objective at hand.

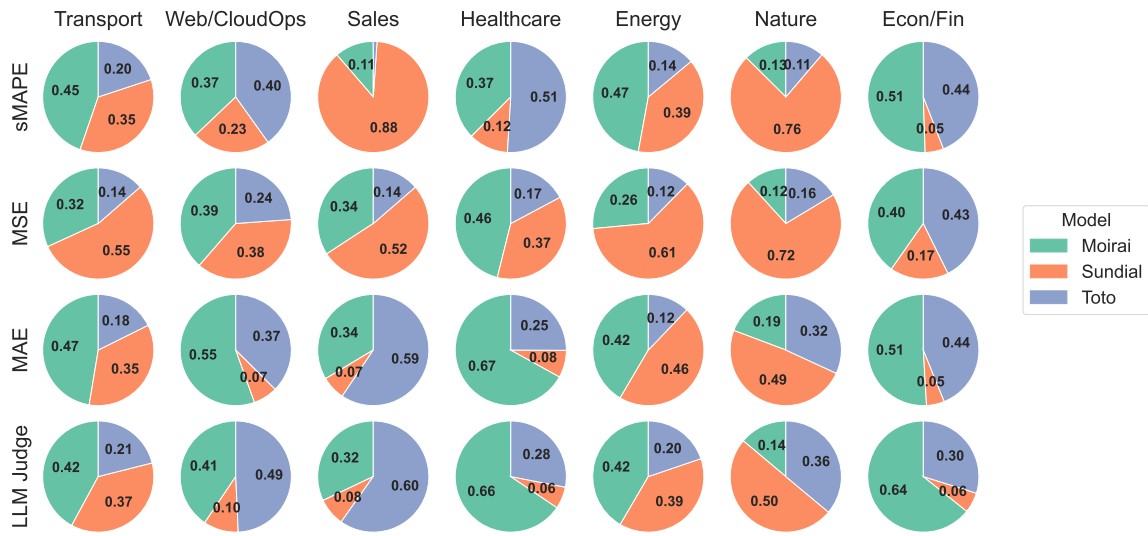

Figure 9: Average ensemble weights assigned to Moirai, Sundial, and Toto across different domains and optimization metrics, along with using the LLM to decide the weights.

### Domain-Specific Weight Adaptation and the Need for LLM-Guided Optimization

Figure 9 reveals crucial insights into ensemble behavior across domains and metrics, demonstrating why automated LLM-guided optimization is essential rather than optional.

The weight distributions expose complex, non-intuitive patterns that would be impossible to manually optimize. In Sales forecasting, Sundial dominates when optimizing for sMAPE (88% weight) but drops to just 11% under MAE optimization—a complete reversal that reflects fundamentally different error characteristics. Healthcare shows similar volatility: Toto receives 51% weight under sMAPE but only 8% under MAE, while Moirai jumps from 37% to 67%. These dramatic shifts aren't random—they reflect each model's architectural strengths interacting with domain-specific patterns and metric sensitivities.

Critically, the "LLM Judge" row demonstrates that our system discovers weight configurations that often differ from any single metric optimization. For Transport, the LLM assigns balanced weights (42/37/21%) that diverge from MSE's Sundial-heavy distribution (32/55/14%), suggesting the LLM identifies complementary model behaviors that single metrics miss. In Web/CloudOps, the LLM's 41/10/49% split represents a unique configuration not achieved by any traditional metric, indicating it captures domain nuances beyond simple error minimization.

**This complexity makes the case for LLM-guided optimization compelling**: with 7 domains × 4 metrics × 3 models = 84 weight parameters to optimize, and non-linear interactions between domain characteristics and model architectures, manual or grid-search approaches become intractable. The LLM judge synthesizes multiple signals—error patterns, temporal characteristics, and cross-validation stability—to discover optimal configurations that pure metric optimization misses. Without this intelligent guidance, practitioners would need deep expertise in each model's architecture and extensive domain knowledge to manually tune weights, making ensemble methods impractical for real-world deployment.

### H.3 CROSS-VALIDATION FOLDS AND OPTIMIZATION METRIC

Figure 10 reports the average ensemble weights across all datasets for Moirai, Sundial, and Toto across different cross-validation folds $T_0, \ldots T_{24}$ when optimizing for either sMAPE, MSE, or MAE. Because ensemble weights are computed independently for each fold, the figure illustrates how each model's importance shifts with respect to the fold and optimization metric.

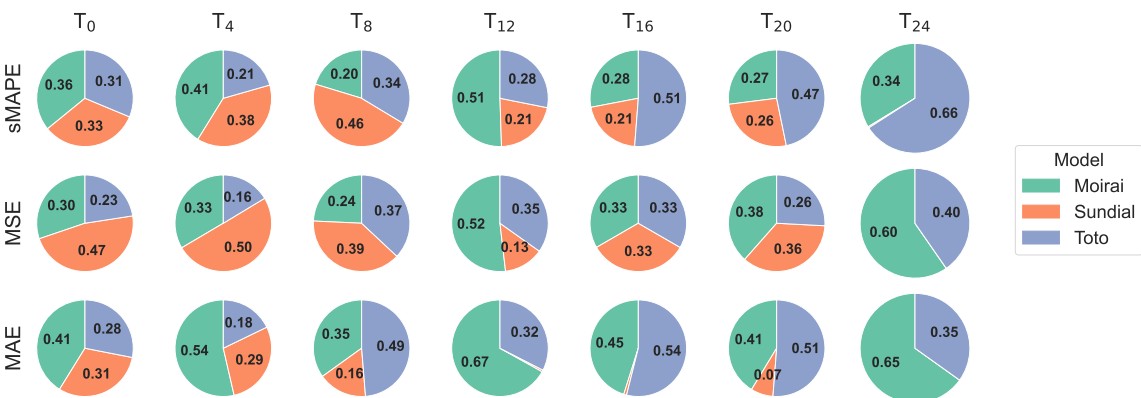

Figure 10: Average ensemble weights assigned to Moirai, Sundial, and Toto across different cross-validation folds when optimizing for different metrics.

Under sMAPE optimization, the weights begin relatively balanced and Toto receives increasingly greater weights with later folds. For MSE, Sundial begins with the most weight and receives less weight with later folds. When optimizing for MAE, the weights also begin relatively even, and Moirai and Toto progressively dominate the ensemble. These patterns highlight how the ensemble adjusts to the data in each batch. Each model receives greater weights depending on the fold's data and the optimization metric — no single model dominates across all folds and metrics. The ability for the ensemble to adjust its weights in response to the fold's data and optimization metric further demonstrates its strength and robustness.

