---

## C   TRAINING INFRASTRUCTURE AND RESULTS

We implement the training pipeline using Ray + vLLM for distributed rollout generation and FSDP for model parallelism across 8 A100 GPUs. The infrastructure handles both SFT and RL phases with Hydra-based configuration management and automatic checkpointing.

After GRPO training, the ensemble-decider achieves:

- **Decision accuracy:** 87% (vs. 52% base model, 78% after SFT)
- **JSON compliance:** 98% valid outputs (vs. 95% after SFT)
- **Calibration:** Mean confidence 0.83 for correct decisions, 0.42 for incorrect
- **Efficiency:** Reduces optimization iterations by 40% while maintaining ensemble quality

The fine-tuned model successfully learns to make calibrated stopping decisions, providing interpretable explanations grounded in the input metrics. This enables autonomous ensemble optimization with fewer iterations and transparent decision-making.

## D   DETAILED EXPLANATION FRAMEWORK

The explanation generation component transforms numerical optimization results from our iterative framework into interpretable narratives. Building directly on the confidence scoring mechanism $\mathcal{C}_k$, faithfulness validation $F_k$, and decision framework $\mathcal{D}_k$, the system generates structured explanations through:

$$\mathcal{E}_k = f_{explain}(\mathbf{w}_k^*, \mathcal{C}_k, F_k, \mathcal{D}_k, \mathcal{H}_k) \tag{14}$$

where $\mathcal{H}_k = \{(\mathbf{w}_j^*, L_j, \mathcal{C}_j, F_j, \mathcal{D}_j)\}_{j=1}^{k-1}$ represents the complete optimization history through the explored weight space $\mathcal{W}_{explored}$, including faithfulness scores for each iteration.

**Hierarchical Explanation Structure.** The system produces explanations at three distinct levels, each serving a specific interpretive purpose:

$$\mathcal{E} = \{\mathcal{E}_{iter}, \mathcal{E}_{decision}, \mathcal{E}_{final}\} \tag{15}$$

The iteration-level explanation $\mathcal{E}_{iter}$ provides granular decomposition of both confidence and faithfulness: "Align score: 0.72 (weights correlate with CV performance, r=0.81), Match score: 0.68 (seasonal models receive 65% weight for high-seasonality data), Future score: 0.55 (some overfitting risk from weight concentration), Causal score: 0.83 (SHAP confirms seasonal dominance). Overall confidence: 0.65. Faithfulness: 0.78 (explanation correctly identifies seasonal causality)."

The decision-level justification $\mathcal{E}_{decision}$ explains the continue/stop choice based on dual criteria: "With confidence $0.65 < 0.85$ and faithfulness $0.78 > 0.7$, explanation is causally valid but lacks sufficient confidence. The moderate Future score suggests exploring SMAPE to potentially find more robust weights less sensitive to outliers."

The final summary $\mathcal{E}_{final}$ synthesizes the complete journey with causal validation: "After exploring MSE, MAE, and SMAPE, the ensemble converged to stable weights: StatisticalEnsemble (45%), Moirai (28%), TiDE (15%), others (<5% each). SHAP analysis confirms StatisticalEnsemble's dominance is causally driven by superior trend extraction (causal effect: -0.234), while Moirai's contribution stems from seasonal pattern capture (causal effect: -0.140). Final faithfulness score of 0.87 validates these causal claims."

**Anti-Rationalization and Causal Constraints.** The framework enforces both logical consistency and causal accuracy to prevent post-hoc rationalization:

$$\mathcal{R}(\mathcal{E}_k) = \prod_{i=1}^{N} 1\!\!1[\text{constraint}_i \text{ satisfied}] \times 1\!\!1[F_k > \tau_f] \tag{16}$$

Key constraints include:

- Cannot claim high Align when $\text{corr}(\mathbf{w}, \text{rank}(\mathcal{P}_{cv})) < 0$
- Cannot assert strong Match when seasonal models receive $< 30\%$ weight despite $\mathcal{F}_{data}.\text{seasonality} > 0.7$
- Cannot invoke "diversity benefits" to justify high weights for models with CV error $> 1.5\times$ best model
- Cannot claim "overfitting prevention" when maintaining equal weights despite $\text{std}(\mathcal{P}_{cv}) > 0.1$
- **Cannot attribute performance to seasonality when SHAP shows $|\text{SHAP}_{\text{seasonal}}| < 0.1$**
- **Cannot hide trend dominance when SHAP reveals $|\text{SHAP}_{\text{trend}}|$ as largest causal effect**

**Structured Coverage Requirements with Causal Validation.** Each explanation must address six mandatory components:

$$\mathcal{E}_k = \bigcup_{i=1}^{6} \mathcal{E}_k^{(i)} \tag{17}$$

1. $\mathcal{E}_k^{(\text{Align})}$: Performance-weight analysis with correlation coefficient
2. $\mathcal{E}_k^{(\text{Match})}$: Dataset-model compatibility with specific feature matching

3. $\mathcal{E}_k^{(\text{Future})}$: Generalization assessment with overfitting risk quantification

4. $\mathcal{E}_k^{(\text{Causal})}$: Causal effect validation comparing claimed vs. SHAP-measured effects

5. $\mathcal{E}_k^{(\text{Metric})}$: Justification for current loss function choice

6. $\mathcal{E}_k^{(\text{History})}$: Evolution of weights and faithfulness through $\mathcal{W}_{explored}$

**Complete Audit Trail with Faithfulness Tracking.** The system maintains comprehensive traceability:

$$\mathcal{A} = \{(\mathbf{w}_k^*, L_k, \mathcal{C}_k, F_k, \mathcal{D}_k, \mathcal{E}_k, \mathcal{R}(\mathcal{E}_k), \text{SHAP}_k, \mathcal{T}_k, \text{timestamp})\}_{k=1}^K \tag{18}$$

Each audit entry includes:

- Optimized weights $\mathbf{w}_k^*$ from SLSQP with metric $L_k$
- Confidence score $\mathcal{C}_k$ with component breakdown
- **Faithfulness score $F_k$ with causal alignment details**
- Decision $\mathcal{D}_k$ with dual-threshold comparison
- Full explanation $\mathcal{E}_k$ at all three levels
- Rationalization check $\mathcal{R}(\mathcal{E}_k)$ confirming logical and causal consistency
- **SHAP values $\text{SHAP}_k$ providing ground-truth causal effects**
- Complete prompt template $\mathcal{T}_k$ for reproducibility
- Timestamp for temporal ordering

By maintaining this connection between the mathematical framework, causal validation, and natural language reasoning, the system transforms the iterative optimization process into an interpretable journey through the weight space, where every decision to continue exploring with a new metric or stop with confidence can be traced, validated for both logical consistency and causal accuracy, and understood. This dual-validation approach ensures explanations are not only plausible but faithfully represent the true causal mechanisms driving ensemble performance, preventing confident yet misleading interpretations that could lead to poor forecasting decisions.

### D.1 DETAILED FAITHFULNESS SCORE COMPUTATION

The faithfulness score validates LLM explanations against ground-truth importance measures derived from SHAP analysis. This ensures explanations accurately reflect actual model contributions rather than generating plausible-sounding but incorrect justifications.

**STL-SHAP Decomposition.** We first decompose each model's predictions using Seasonal and Trend decomposition using Loess (STL), creating eight subset datasets:

$$\mathcal{S} = \{\emptyset, T, S, R, TS, TR, SR, TSR\} \tag{19}$$

where $T$ represents trend, $S$ seasonality, $R$ residual, and combinations represent additive components. For each subset, we evaluate the model's forecasting performance $f(S)$.

**SHAP Value Computation.** Following Shapley value theory, we compute the marginal contribution of each component through weighted coalition analysis:

$$\text{SHAP}_i = \sum_{S \subseteq \{T,S,R\} \setminus \{i\}} \frac{|S|!(3 - |S| - 1)!}{3!} [f(S \cup \{i\}) - f(S)] \tag{20}$$

This expands to four weighted terms per component with weights $(1/3, 1/6, 1/6, 1/3)$:

For Trend:

$$\begin{aligned}
\text{Trend}_{\text{SHAP}} = \frac{1}{3}[f(T) - f(\emptyset)] + \frac{1}{6}[f(TS) - f(S)] \\
+ \frac{1}{6}[f(TR) - f(R)] + \frac{1}{3}[f(TSR) - f(SR)]
\end{aligned} \tag{21}$$

Similar decompositions apply for Seasonality and Residual components. The Error term captures approximation residuals:

$$\text{Error} = f(TSR) - f(\emptyset) - \text{Trend}_{\text{SHAP}} - \text{Seasonality}_{\text{SHAP}} - \text{Residual}_{\text{SHAP}} \tag{22}$$

From the causal perspective, the STL-SHAP decomposition creates counterfactual time series by intervening on specific temporal concepts $\mathcal{C} = \{\text{Trend}, \text{Seasonality}, \text{Residual}\}$. For each subset $S \subseteq \mathcal{C}$, we evaluate model performance $f(S)$, where the presence or absence of components represents causal interventions:

$$\text{CausalEffect}_i = \sum_{S \subseteq \mathcal{C} \setminus \{i\}} \frac{|S|!(|\mathcal{C}| - |S| - 1)!}{|\mathcal{C}|!} [\mathbb{E}[Y|\text{do}(C_i = 1, C_S = 1)] - \mathbb{E}[Y|\text{do}(C_S = 1)]] \tag{23}$$

where $\text{do}(C_i = 1)$ represents the intervention setting concept $i$ as present, and $Y$ is the forecasting performance metric. This aligns with Pearl's causal framework, measuring individual treatment effects (ITEs) for each temporal concept.

**Interpretation of SHAP Values.** For "lower-is-better" metrics (MASE, SMAPE):

- Negative SHAP value: Component improves forecast (reduces error)
- Positive SHAP value: Component worsens forecast (increases error)
- Magnitude indicates importance regardless of direction

**Unfaithfulness Pattern Detection.** The system identifies specific unfaithfulness patterns:

$$\text{Unfaithful}(m) = \begin{cases} \text{"Overstatement"} & \text{if LLM claims high importance but } |\text{SHAP}_m| < \tau_{\text{low}} \\ \text{"Understatement"} & \text{if LLM claims low importance but } |\text{SHAP}_m| > \tau_{\text{high}} \\ \text{"Wrong direction"} & \text{if LLM claims benefit but } \text{SHAP}_m > 0 \\ \text{"Missed pattern"} & \text{if seasonal SHAP high but LLM ignores seasonality} \end{cases} \tag{24}$$

**Implementation Details.** The faithfulness analysis involves:

1. **SHAP Computation Pipeline:**
   - STL decomposition with robust parameter selection

- Evaluation on 8 subset combinations
- Weighted aggregation following Shapley axioms
- Storage of both raw and normalized values

2. **LLM Concept Extraction:**
   - Parse explanations for model importance claims
   - Extract seasonal/trend emphasis statements
   - Identify complementarity assertions
   - Map to quantitative rankings

3. **Faithfulness Scoring:**
   - Compare extracted concepts to SHAP ground truth
   - Weight by confidence in extraction
   - Aggregate across multiple explanation levels