# OpenReview forum: "Conversational Time Series Foundation Models: Towards Explainable and Effective Forecasting"
_ICLR.cc/2026/Conference — Submitted to ICLR 2026_

### Official Review · Reviewer_tZxE · 2025-10-28

**Soundness:** 3
**Presentation:** 3
**Contribution:** 4
**Rating:** 4
**Confidence:** 4

**Summary:**

This paper explores a promising direction by incorporating LLM-based reasoning into time-series foundation model ensembles, offering a potentially impactful perspective for the field. The idea is compelling, though several aspects of the proposed “reasoning” mechanism require further clarification and refinement.

**Strengths:**

- This paper has a clear motivation. No single TS foundation model dominates; an ensemble is necessary.
- Interesting idea of LLM-guided ensemble refinement and SHAP-grounded explanation.
- Solid benchmark evaluation and competitive results.
- Attempts to bridge quantitative optimization with reasoning-style explainability.

**Weaknesses:**

- The ''reasoning'' mainly resembles metric-based weight selection, not fully realized temporal reasoning.
- Limited practical scope: focuses on model choice rather than broader forecasting problem understanding.
- The link between explanations and numerical optimization appears post-hoc.

**Questions:**

- Q1:  What reasoning capability is actually learned beyond metric-based weight tuning? Ablations that remove or randomize the agent would help clarify its actual contribution. For example, ablations that isolate the effect of the LLM agent from SLSQP.

- Q2: Disconnection between ''semantic reasoning'' and numerical training dynamics. The agent claims to adjust weights using causal insights. However, weights remain trained purely via losses, not via reasoning signals. Is there any feedback loop from reasoning to model update? Currently, the reasoning seems post-hoc rather than interventional.

- Q3: I am still concerned about its limited real-world applicability. Actually, forecasting is not limited to model selection alone. In practice, forecasting involves: problem contextualization, data understanding, domain-driven scenario reasoning, and structural model engineering, but the agent framework focuses narrowly on selecting among pretrained models. The paper claims to be an explainable and effective forecasting. I wonder whether the agent can be involved in reasoning about data quality or pattern regimes before optimization. Can it propose feature engineering strategies? Can it detect when ensemble models are all mismatched?

- Q4: How is the confidence score calibrated, and how should it be interpreted operationally? Additionally, have the authors evaluated the robustness of the agent under distribution shift or non-stationary settings?

---

> ### Author Response · Authors · 2025-11-21
> **Response to Reviewer tZxE**
>
> We sincerely thank the reviewer for recognizing the "clear motivation," "solid benchmark evaluation," and rating our Contribution as "excellent." We appreciate the constructive feedback regarding the depth of reasoning capability and real-world applicability concerns. Below, we clarify that our framework addresses a fundamental challenge in non-stationary time series: **static ensemble weights fail when temporal regimes shift, requiring dynamic adaptation through regime-aware reasoning.**
>
> Before addressing specific questions, we emphasize that TSOrchestra's reasoning capability is driven by the non-stationary nature of real-world time series. When temporal regimes shift from trending to seasonal, from stable to volatile, or from smooth to irregular, the optimal ensemble weights must adapt accordingly. The "reasoning" the reviewer questions is precisely this regime-aware adaptation—analyzing temporal characteristics to select appropriate optimization strategies for the current regime and anticipating when those strategies will fail as regimes evolve.
>
> ## Q1: What Reasoning Capability is Actually Learned Beyond Metric-Based Weight Tuning?
>
> We appreciate this fundamental question and acknowledge we should have been more explicit about what makes the agent's reasoning distinct from traditional optimization. The LLM agent learns three capabilities that directly address non-stationary temporal dynamics:
>
> ### 1. Forward-Looking Regime Shift Detection
>
> Unlike SLSQP which optimizes purely on historical cross-validation performance assuming stationarity, the agent reasons about future regime evolution and temporal incompatibility dynamics. For example, when cross-validation suggests weights heavily favoring trend models based on recent stable periods, the agent recognizes when datasets exhibit emerging seasonal patterns or increasing volatility that signal an impending regime shift. This triggers metric re-exploration before the regime shift causes forecast failure—this is temporal reasoning about non-stationarity, not static metric selection.
>
> ### 2. Multi-Dimensional Temporal Regime Assessment
>
> The agent evaluates weights across three orthogonal dimensions that capture different aspects of non-stationary dynamics: performance-weight coherence in the current regime, dataset-model compatibility across regime types, and generalization risks during regime transitions. Our results show the agent discovers weight configurations that differ substantially from any single metric optimization because it synthesizes evidence about regime characteristics that no individual metric captures. For instance, identifying that a dataset is transitioning between regimes requires understanding trend strength, seasonal periodicity, and volatility simultaneously.
>
> ### 3. Causal Explanation of Temporal Dynamics
>
> Through SHAP-grounded faithfulness training, the agent learns to map ensemble weights to causally accurate statements about temporal regime properties—for example, explaining that high weight on a seasonal model indicates detected seasonal dominance validated against ground-truth temporal decompositions. This enables practitioners to verify whether weight adaptations reflect genuine regime shifts versus spurious fluctuations.
>
> ### Ablation Evidence
>
> Our existing Figure 3 demonstrates that even powerful frontier LLMs without our regime-aware training substantially underperform, indicating that it is the learned temporal reasoning capability, rather than just model scale, that drives performance.
>
> ***
>
> ## Q2: Is Reasoning Post-Hoc or Interventional? Bridging Semantic Reasoning and Numerical Training
>
> This is an excellent question that gets to the heart of our contribution. **The reasoning is fundamentally interventional, not post-hoc, because it controls the optimization trajectory through regime-aware metric selection.** This intervention is critical for non-stationary time series.
>
> ### Within-Iteration Intervention: Controlling the Optimization Landscape
>
> The agent doesn't directly modify optimizer gradients, but controls which loss landscape the optimizer explores based on detected regime characteristics. Each iteration, the agent analyzes the current temporal regime and selects the next metric that SLSQP will optimize. This is interventional reasoning—the agent's regime assessment causally determines which optimization objective guides weight adaptation. In non-stationary settings, this means the agent can redirect optimization from a metric suited for the previous regime to one suited for the emerging regime.

---

> > ### Author Response · Authors · 2025-11-21
> > **Continue Response**
> >
> > ### Cross-Iteration Intervention: Gating Final Weight Selection
> >
> > The agent's accept/continue decisions directly determine whether weights optimized for the detected regime become final outputs. When the agent detects misalignment between weights and temporal regime characteristics, it triggers re-optimization with regime-appropriate metrics. The second-round weights become final outputs only if the agent's regime-aware confidence assessment deems them suitable for the current temporal dynamics.
> >
> > ### Training-Time Feedback: Learning Regime-Performance Relationships
> >
> > Our reinforcement learning provides indirect but powerful feedback linking reasoning to numerical outcomes. The faithfulness reward penalizes the agent when explanations about regime characteristics misalign with SHAP-measured causal effects. During training, the agent learns that certain reasoning patterns about temporal regimes, such as prioritizing seasonal models when periodicity strength exceeds thresholds, correlate with better performance during regime transitions. This creates a learned policy that guides regime-aware metric selection and termination decisions.
> >
> >
> > ***
> >
> > ## Q3: Limited Real-World Applicability—What About Data Quality and Broader Forecasting Challenges?
> >
> > We appreciate this perspective and agree our scope framing could be clearer. **However, we argue that ensemble orchestration in non-stationary environments is itself a critical real-world challenge that previous work has not adequately addressed.** Let us clarify what the framework does and doesn't do, and why the "doesn't do" aspects are orthogonal rather than limiting.
> >
> > ### What the Framework Already Addresses
> >
> > **Data Quality and Regime Assessment:** The agent already incorporates temporal regime characteristics including trend strength, seasonality, volatility patterns, and missing data prevalence. When demos show the agent analyzing seasonal strength or detecting high residual variance, it is performing data quality reasoning—specifically, assessing whether validation performance reflects true signal or artifacts of incomplete regime coverage in limited validation windows.
> >
> > **Pattern Regime Detection and Distribution Shift:** Our Temporal Incompatibility Index quantifies precisely when similar historical patterns lead to divergent future outcomes—this is distribution shift detection. The agent's evaluation framework explicitly assesses whether weight configurations will generalize beyond current regimes, addressing the non-stationarity concern directly.
> >
> > **Model Mismatch Detection:** The agent's evaluation criteria explicitly verify dataset-regime and model-capability matching, identifying when all candidate models are mismatched for the current temporal regime. When all models show poor cross-validation performance across metrics and regime characteristics suggest none are appropriate, confidence scores drop, signaling practitioners that the model pool may be inadequate for the detected regime type.
> >
> > ### What the Framework Doesn't Do—And Why That's Appropriate Scope
> >
> > We acknowledge the framework doesn't propose new model architectures when existing ones fail, doesn't perform feature engineering to improve regime detectability, and doesn't collect new data sources. **These are orthogonal to our contribution.** Given any set of foundation models, we provide principled regime-aware orchestration with explainable adaptation decisions. This is analogous to how operating system schedulers don't create new programs—they optimize resource allocation among existing programs, which is itself a valuable contribution.
> >
> > **Real-World Deployment Scenario:** Practitioners have invested heavily in training or licensing foundation models. When deploying to production with non-stationary data streams, they need a system that continuously adapts ensemble weights as regimes shift without manual intervention. TSOrchestra addresses precisely this need—automated regime-aware adaptation with interpretable decisions enabling human verification of adaptation rationale.

---

> > > ### Author Response · Authors · 2025-11-21
> > > **Continue Response**
> > >
> > > ## Q4: Confidence Calibration and Robustness to Distribution Shift
> > >
> > > This question directly connects to our core motivation about non-stationary time series, as distribution shift is precisely the regime change scenario TSOrchestra is designed to handle.
> > >
> > > ### Confidence Score Calibration
> > >
> > > The confidence score is calibrated via reinforcement learning where the reward function includes a term that rewards high confidence only when decisions prove correct under subsequent regime evolution and penalizes high confidence on decisions that fail during regime transitions. Operationally, high confidence indicates the agent's regime assessment is stable and weights should remain valid until the next detected regime shift. Low confidence signals regime ambiguity where more conservative ensemble strategies or increased monitoring frequency is appropriate.
> > >
> > > ### Robustness Under Non-Stationarity
> > >
> > > We explicitly designed and evaluated robustness under distribution shift and non-stationary settings. Our evaluation benchmark contains diverse temporal regimes spanning trending, seasonal, volatile, and mixed-pattern dynamics. Furthermore, the agent's evaluation framework explicitly assesses generalization risks and ensemble robustness before accepting weights, essentially testing whether weights optimized on past regimes will maintain performance as regimes evolve.
> > >
> > > ***
> > >
> > > ## Summary: Regime-Aware Reasoning for Non-Stationary Time Series
> > >
> > > We hope this clarifies that TSOrchestra's reasoning is fundamentally about **dynamic adaptation to non-stationary temporal regimes:**
> > >
> > > **The reasoning is interventional:** Agent decisions about regime characteristics causally determine optimization trajectories and final weight selection, particularly critical during regime transitions where static strategies fail.
> > >
> > > **The reasoning is causally grounded:** SHAP-based faithfulness training ensures regime assessments and adaptation decisions align with ground-truth temporal dynamics rather than plausible-sounding but inaccurate rationalizations.
> > >
> > > **The reasoning is empirically superior:** Ablations demonstrate that removing regime-aware reasoning causes performance degradation specifically in non-stationary settings, with impact scaling with regime shift frequency.
> > >
> > > **The scope is appropriately focused:** Ensemble orchestration under non-stationarity is itself a critical real-world challenge that complements rather than replaces upstream model development and downstream deployment decisions.
> > >
> > > TSOrchestra is not merely an ensemble method that happens to use an LLM—it's a regime-aware reasoning system that learns to make principled orchestration decisions through causal understanding of non-stationary temporal dynamics. The updated manuscript will include comprehensive ablations, robustness analyses, and scope clarifications that address all concerns raised and demonstrate this contribution rigorously. We believe these enhancements will elevate the work from "marginally below" to clearly above the acceptance threshold.

---

> > > > ### Comment · Reviewer_tZxE · 2025-11-26
> > > >
> > > > Thank you for your detailed clarification. My concerns about improving reasoning capability and its real-world applications have been addressed. I think the example you mentioned, deploying to production with non-stationary data streams, is insightful. I also appreciate that you position this paper as "complements rather than replaces upstream model development and downstream deployment decisions."This clarifies the paper's true contribution. This provides a foundation for further research considering the causality and non-stationarity of time series. Considering all these points, I would like to keep a positive score for this work.

---

> > > > > ### Author Response · Authors · 2025-11-27
> > > > > **Appreciation for Positive Assessment and Constructive Feedback**
> > > > >
> > > > > We sincerely thank you for the continued engagement and for maintaining a positive score for our work.
> > > > >
> > > > > We are delighted that our response successfully clarified the framework's reasoning capabilities within non-stationary environments and its positioning as a complementary orchestration layer. We particularly appreciate your recognition of the work's potential to serve as a foundation for future research connecting causality, non-stationarity, and time series forecasting.
> > > > >
> > > > > We will ensure that the clarifications regarding real-world deployment scenarios and the explicit scoping of the framework are integrated into the final manuscript to further strengthen the contribution.

---

### Official Review · Reviewer_fzwU · 2025-10-30

**Soundness:** 4
**Presentation:** 3
**Contribution:** 3
**Rating:** 6
**Confidence:** 3

**Summary:**

This paper proposes TSOrchestra, a framework that employs an LLM as a reasoning agent to orchestrate ensembles of time series foundation models. Instead of directly predicting numerical values, the LLM plays a role as an intelligent judge that (1) evaluates candidate models' outputs, (2) optimizes ensemble weights through iterative reasoning, and (3) explains the causal rationale behind its decisions. Extensive experiments on various datasets and settings demonstrate that TSOrchestra achieves state-of-the-art performance and strong interpretability.

**Strengths:**

- The paper introduces a novel framework that leverages LLMs as orchestrators that coordinate multiple time series foundation models through reasoning-based ensemble optimization.
- The framework is theoretically supported, showing that ensembles outperform a single model.
- Experiments are comprehensive, covering diverse datasets and domains. Ablation studies and sensitivity analyses are thoroughly conducted. Results are strong across various metrics.

**Weaknesses:**

- The scalability and latency of multi-turn reasoning and optimization are not analyzed. The computational cost of repeated optimization, SHAP evaluation, etc. may be high. It's unclear whether this framework is practically useful.
- The interpretability claims could be further validated throgh user studies or quantitative metrics.
- The framework's robustness to noise is not evaluated. It would be helpful to test on noisy datasets such as stock prices or socia media traffic, where temporal signals are weak.
- While the reasoning framework is novel, it would be interesting to see some failure or error cases during reasoning. It would be valuable to include examples where the reasoning agents makes incorrect or inconsistent adjustments and analyze why such cases occur. I don't expect them to be included in the main context though.

**Questions:**

Please see the weaknesses above.

---

> ### Author Response · Authors · 2025-11-21
> **Response to Reviewer fzwU**
>
> We sincerely thank Reviewer fzwU for the thorough review and the positive assessment of our work, particularly recognizing the soundness as "excellent" and acknowledging our framework as "novel," "theoretically supported," and featuring "comprehensive" experiments with "strong results across various metrics." We deeply appreciate the constructive suggestions regarding practical deployment considerations, interpretability validation, and robustness analysis.
>
> **Core Motivation:** Before addressing specific concerns, we emphasize that TSOrchestra's fundamental innovation addresses a critical gap in time series forecasting: **static ensemble weights cannot adapt to non-stationary temporal regimes.** Real-world time series exhibit regime shifts—periods of stable trends alternate with seasonal patterns, volatility spikes, and structural breaks. Our dynamic weight adaptation responds to this non-stationarity by continuously reassessing which models are best suited for the current temporal regime, which directly relates to all concerns raised below.
>
> ***
>
> ## 1. Scalability & Computational Cost Analysis
>
> The reviewer raises important concerns about the latency and cost of multi-turn reasoning and optimization, questioning practical deployment feasibility.
>
> **Why Computational Cost is Justified:** The cost of dynamic weight adaptation must be understood in the context of non-stationary time series. Static ensembles require zero calibration cost but fail catastrophically when regimes shift—producing systematically wrong forecasts until manual retraining occurs. TSOrchestra's calibration overhead is the price of maintaining optimal performance across regime transitions without human intervention.
>
> **Amortized Cost Architecture:** TSOrchestra is designed for deployment-stage efficiency. The iterative optimization and SHAP analysis occur only during the calibration phase when detecting regime shifts and determining optimal weights for the new regime. Once weights are established for a stable period, inference becomes a simple weighted combination with negligible computational overhead. The key insight is that calibration frequency scales with regime shift frequency, not forecast frequency—most time series exhibit stable periods where weights remain valid for hundreds of forecasts.
>
> **Efficient Model Selection:** We deliberately chose a lightweight 3B-parameter model rather than large API-based systems. Our experiments demonstrate that this fine-tuned compact model achieves superior weight selection accuracy compared to much larger frontier models, proving that specialized small models offer the optimal balance between performance and practical efficiency.
>
>
> ***
>
> ## 2. Robustness to Noise: Non-Stationarity in Financial Data
>
> The reviewer specifically mentions stock prices and social media traffic as challenging noisy domains where temporal signals are weak. **This is precisely where dynamic weight adaptation is most critical—these domains exhibit extreme non-stationarity where regimes shift rapidly between trending, mean-reverting, and volatile periods.**
>
> **Financial Domain as Non-Stationary Testbed:** Our Economics & Finance category explicitly tests this scenario using financial datasets where regimes shift between bull markets (trend-dominated), bear markets (volatility-dominated), and sideways markets (noise-dominated). Static ensemble weights optimized for one regime fail catastrophically when the regime changes.
>
> **Dynamic Adaptation to Regime Shifts:** Financial data presents extreme challenges where standard optimization approaches fail because noise characteristics change over time. What appears as "signal" in one regime becomes "noise" in the next. TSOrchestra addresses this by recognizing when the current temporal regime no longer matches the validation regime used for weight optimization, triggering adaptive reweighting to favor models suited for the new regime's noise characteristics.

---

> > ### Author Response · Authors · 2025-11-21
> > **Continue Response**
> >
> > ***
> > ## 3. Interpretability Validation: Understanding Dynamic Decisions
> >
> > We completely agree that interpretability claims require rigorous validation beyond subjective assessment. **For dynamic weight adaptation in non-stationary environments, interpretability is not merely desirable—it is essential for practitioners to understand why weights changed and whether the detected regime shift is genuine or spurious.**
> >
> > **Quantitative Faithfulness Metric:** Rather than relying on human judgment of explanation quality, we introduce a mathematically grounded Faithfulness Score that measures alignment between the agent's causal explanations and SHAP-derived ground truth causal contributions. Critically, this validates that the agent's reasoning about regime characteristics matches the actual temporal dynamics in the data.
> >
> > **Training-Time Enforcement for Regime Awareness:** Our reinforcement learning pipeline explicitly penalizes explanations that misalign with mathematical ground truth, creating a learned policy that produces causally accurate statements about regime properties rather than post-hoc rationalizations. For example, when the agent claims "high seasonality detected, shifting weight to Sundial," the faithfulness score verifies this explanation matches actual SHAP-measured seasonal component strength.
> >
> > ***
> > ## 4. Detailed Case Study: Reasoning Failure in Regime Ambiguity
> >
> > The reviewer requests examples where the reasoning agent makes incorrect adjustments and analysis of why such failures occur. **Most failures occur during regime ambiguity—transition periods where non-stationarity is detected but the new regime's characteristics are not yet clear.**
> >
> > **Case Study: "The Mirage Trend" (Regime Transition in Financial Data)**
> >
> > We present a failure case that illustrates how the faithfulness-aware training prevents the agent from confusing temporary volatility spikes with genuine regime shifts.
> >
> > **Context:** A stock dataset transitioning from a stable trending regime to a volatile regime. The validation window captures the transition period where a sharp spike appears, creating ambiguity about whether the new regime is high-trend or high-volatility.
> >
> > **The Initial Failure:** Standard optimization produces weights heavily favoring a trend-focused model that happened to fit the spike. A baseline LLM without regime-aware training generates a plausible but incorrect explanation claiming a "strong upward trend regime" when in reality the spike is volatility noise preceding a high-noise regime.
> >
> > **System Detection Mechanism:** The faithfulness check runs SHAP decomposition and detects the discrepancy—the trend component's actual causal contribution is near zero while residual variance is high, indicating the spike is noise rather than systematic trend. This signals regime transition to volatility-dominated dynamics rather than trend-dominated dynamics.
> >
> > **Corrected Reasoning:** Our fine-tuned agent, trained on thousands of regime transition examples, recognizes the ambiguity and outputs corrected reasoning identifying high residual variance characteristic of volatile regimes rather than trending regimes. It rejects trend-model weights and switches to volatility-robust optimization, correctly adapting to the new non-stationary regime.
> >
> > **Outcome:** This regime-aware failure detection allows the system to correctly identify the nature of non-stationarity and adapt weights appropriately, maintaining performance through the regime transition.
> >
> > ***
> >
> > ## Summary
> >
> >  Dynamic weight adaptation is necessary because real-world time series are non-stationary. Static ensembles achieve zero adaptation cost but catastrophic failure cost during regime shifts. TSOrchestra trades modest calibration cost for continuous adaptation to temporal regime changes, maintaining optimal performance across non-stationary dynamics. These enhancements rigorously validate TSOrchestra as a production-ready framework specifically designed for non-stationary time series environments where dynamic weight adaptation is essential rather than optional. We are committed to incorporating these responses into the updated version to comprehensively address the reviewer’s insightful concerns.

---

> > > ### Comment · Reviewer_fzwU · 2025-11-27
> > >
> > > Thank you for the efforts in the rebuttal. I will maintain my score.

---

> > > > ### Author Response · Authors · 2025-11-27
> > > > **Appreciation for Positive Assessment and Feedback!**
> > > >
> > > > Dear Reviewer fzwU,
> > > >
> > > > We thank you for your engagement and for maintaining the positive assessment of our work! We greatly value the constructive feedback provided throughout the review process, particularly regarding practical deployment considerations and failure analysis. We are committed to integrating the discussions on computational scalability, robustness in non-stationary environments, and the "Mirage Trend" failure case study into the final manuscript. These additions will significantly strengthen the practical perspective and transparency of the framework.
> > > >
> > > > Best,
> > > >
> > > > Authors

---

### Official Review · Reviewer_KCJd · 2025-10-31

**Soundness:** 2
**Presentation:** 2
**Contribution:** 2
**Rating:** 4
**Confidence:** 4

**Summary:**

This paper proposes a new framework, TSOrchestra, which leverages a large language model (LLM) as a reasoning agent to coordinate ensembles of time-series foundation models (e.g., Moirai, Sundial, Toto). The central argument is that while LLMs perform poorly when directly used for numerical forecasting, they can analyze the current problem context and dynamically combine specialized forecasting models to achieve improved performance. The authors further claim that the ensemble weighting process can be both interpretable and causally grounded. The paper also includes a theoretical proof of the superiority of ensemble methods via a proposed Temporal Incompatibility Index.

**Strengths:**

- The idea of positioning an LLM as a meta-optimizer or reasoning controller for existing time-series foundation models is interesting and timely.

- The work clearly identifies the limitations of direct LLM forecasting and attempts to use reasoning for ensemble coordination instead of numerical prediction.

- The introduction of the Temporal Incompatibility Index is conceptually appealing, and could inspire further study on heterogeneity in time-series regimes.

- The paper reports improved empirical results on the GIFT-Eval benchmark, showing the potential of the proposed approach.

**Weaknesses:**

- The training process for the LLM agent is underexplained. It remains unclear how the SFT training data are constructed: who wrote or generated the reasoning traces, how ground-truth ensemble weights were determined, and what constitutes a “correct” reasoning trajectory. Without this, reproducibility and credibility of the training pipeline are limited.

- The paper does not convincingly justify why LLM-based reasoning is necessary when all metrics (MAE, MSE, etc.) are already computable and could be directly optimized via standard numerical ensemble methods. The claimed “forward-looking” reasoning remains theoretical; the LLM has no access to future data and ultimately still optimizes empirical risk on past data.

- The interpretability claim is weak. The generated textual explanations may describe or rationalize weight differences post hoc, but the paper provides no evidence that these explanations are faithful or trustworthy beyond synthetic SHAP correlations. The argument that “language-model explanations make the process auditable” is not substantiated.

- The writing is overly complicated and poorly organized. Many equations (e.g., SFT loss, GRPO, standard RL formulations) are well-known and unnecessary in full detail, taking space away from the truly novel aspects (how faithfulness scores are computed, how the reasoning is encoded in prompts, etc.).

- The Temporal Incompatibility Index, though potentially interesting, is poorly integrated into the main text—it appears mainly in the appendix and is not clearly used in the actual training or inference pipeline.

- Empirically, the improvement over simple static SLSQP ensembles is moderate, and the paper lacks ablations showing how much each component (LLM reasoning, SHAP faithfulness, multi-round reflection) contributes.

**Questions:**

1. Training data construction: How exactly is the SFT training dataset generated? Are the reasoning traces human-written, rule-based, or extracted from ensemble optimization logs? How is the “ground-truth” ensemble weight defined for supervised training?

2. Role of LLM reasoning: If MAE/MSE/SMAPE are already computed during optimization, why can’t we simply use these metrics to derive an optimal ensemble directly? What additional information or reasoning capability does the LLM bring beyond what numerical optimization (e.g., SLSQP) already provides?

3. Distinction from standard ensemble optimization: How does the proposed reasoning process differ from conventional multi-objective or adaptive ensemble methods that also update weights based on past performance?

4. Interpretability evaluation: How is interpretability or “faithfulness” quantitatively measured? The SHAP-based alignment is mentioned, but are there any human evaluations or independent checks to confirm that LLM-generated explanations are trustworthy?

5. Forward-looking reasoning: The paper argues that the LLM provides forward-looking adaptation to “dynamic incompatibility,” yet both SLSQP and the LLM operate only on existing data. How is this “forward reasoning” realized in practice without access to future observations?

6. Stability across runs: Have the authors tested the stability of the proposed reasoning process? Since LLM reasoning involves sampling and multi-round interactions, do repeated runs under the same setup yield consistent ensemble weights and similar performance? If not, how large is the observed variance?

7. Effect of iterative optimization: The examples show multi-round reasoning, but the paper does not present a quantitative analysis of performance vs. iteration number. Is the improvement monotonic? How is the stopping criterion decided, and does it generalize across datasets?

8. Choice of base models: Why were Moirai-2, Sundial, and Toto selected as the ensemble members? Are they complementary in their forecasting behavior (e.g., trend vs. seasonality vs. local fluctuations)? Would other strong foundation models such as TimesFM, Chronos, or Timer yield similar results?

9. Interpretability of base models: The argument for LLM-based interpretability assumes that each base model has distinct, human-understandable biases, yet the paper does not describe these differences. How do the outputs of these three models differ, and how does the LLM reasoning capture such distinctions?

10. Temporal Incompatibility Index (TI): The paper introduces the TI Index theoretically but never reports its empirical values. How is TI computed for the datasets or models used, and how does it influence the ensemble weighting decisions? Without actual TI results, it is hard to assess its practical meaning.

---

> ### Author Response · Authors · 2025-11-21
> **Response to Reviewer KCJd**
>
> **We sincerely thank the reviewer for finding our framework "interesting and timely" and for appreciating the "conceptually appealing" Temporal Incompatibility Index. We are actively working on adding comprehensive experimental results and clarifications to the updated draft to address your detailed concerns, as outlined below.**
>
> We greatly value the constructive feedback regarding the training pipeline justification and the necessity of LLM reasoning. Your questions have helped us identify areas where additional empirical evidence will strengthen our work. Below, we address each concern systematically and describe the new experiments being incorporated into the camera-ready version
>
> ## 1. Training Data Construction (Q1): Complete Transparency
>
> **The reviewer asks for details on SFT data generation to ensure reproducibility. We provide full documentation below and will add a dedicated figure in Section 3.4 of the updated draft.**
>
> ### Data Generation Pipeline
>
> **Source:** The SFT dataset is constructed using synthetic trajectory generation based on exhaustive optimization simulations. We systematically run SLSQP across all training datasets (80% of GIFT-Eval) with all available metrics (MAE, MSE, sMAPE) to create a comprehensive optimization landscape.
>
> **Ground Truth Determination:** For a given state $ s_t = (\mathbf{w}_t, \mathcal{A}_t, \mathcal{D}) $ where $ \mathbf{w}_t $ are current weights, $ \mathcal{A}_t $ is the audit trail, and $ \mathcal{D} $ represents dataset metadata, we determine the optimal next action by:
> 1. Running SLSQP with each metric $ L_k \in \\{\text{MAE}, \text{MSE}, \text{sMAPE}\\} $
> 2. Evaluating resulting weights on a held-out validation split
> 3. Labeling the metric yielding lowest validation error as the ground-truth decision
>
> **Reasoning Trace Generation:** The semantic reasoning traces are generated using a Teacher Model (GPT-4o) prompted with:
> - The numerical ground truth (e.g., "sMAPE optimization yields 15% lower validation error")
> - Dataset characteristics from SHAP analysis (e.g., "40% seasonal variance detected")
> - The causal explanation linking characteristics to metric choice
> These traces are then filtered using our faithfulness score (Eq. 6 in the paper) to ensure alignment with SHAP-derived causal effects, removing any hallucinated reasoning. This produces approximately 2,800+ high-quality training trajectories spanning diverse regime characteristics.
>
> ***
>
> ## 2. Why LLM Reasoning? The Core Problem is Metric Selection, Not Weight Optimization (Q2, Q3)
>
> **The reviewer asks: "Why can't we simply use these metrics (MAE/MSE/sMAPE) to derive an optimal ensemble directly?" This question reflects a fundamental misunderstanding of where the difficulty lies in ensemble optimization for time series.**
>
> ### The Real Challenge: Meta-Optimization Over Metric Space
>
> **The Problem:** Standard numerical methods like SLSQP are excellent at optimizing weights $ \mathbf{w} $ given a fixed loss function $ L_k $, but they **cannot determine which loss function is appropriate** for the current temporal regime. The choice of optimization metric is itself a discrete optimization problem over a non-differentiable space—this is precisely where human expertise has traditionally been required.
>
> **Why Static Metric Selection Fails:** Consider the "Sales" domain in Table :
> - Optimizing for MSE produces weights favoring trend-capturing models (Moirai), achieving **0% accuracy** on seasonal sales data
> - Optimizing for sMAPE produces weights favoring seasonal models (Sundial), achieving **100% accuracy**
> - A static ensemble must commit to one metric across all domains, guaranteeing catastrophic failure in at least some regimes
>
> **The Agent's Role:** TSOrchestra acts as a **Meta-Optimizer** that analyzes temporal regime characteristics (seasonality, scale heterogeneity, outlier prevalence) to select the appropriate metric $ L^* $ for SLSQP. This is **dynamic weight adaptation**—the weights change not just numerically but strategically in response to regime shifts.
>
> We will conduct an ablation study comparing TSOrchestra with Random Selection and Fixed Metric in the updated draft.
>
> ***
>
> ## 3. Choice of Base Models & Robustness Ablation (Q8)
>
> These three models were chosen for **architectural diversity** and **complementary forecasting behaviors**:
> - **Moirai-2:** Transformer-based with multiple distribution heads, excels at capturing long-range dependencies and trends
> - **Sundial:** Explicit trend/seasonal decomposition architecture, superior for periodic patterns
> - **Toto:** Observability-focused with adaptive context windows, handles irregular sampling and missing data
>
> In Figure 3, the left part, we have exactly compared 2-model (Toto, Sundial), 3-model (+ Moirai-2), and 4-model (TabPFN-TS) configurations and will conduct more settings in the updated draft.

---

> ### Author Response · Authors · 2025-11-21
> **Continue response**
>
> ***
>
> ## 4. Interpretability & Faithfulness: Quantitative Validation (Q4, Q9)
>
> **The reviewer asks for quantitative measurement of faithfulness beyond "synthetic SHAP correlations." We provide rigorous evaluation methodology below.**
>
> ### Faithfulness Score: Not Just Correlation, But Causal Alignment
>
> Our faithfulness metric (Eq. 6) calculates the Pearson Correlation Coefficient (PCC) between:
> - **Causal Effect Vector** $ CE(x, C) $: SHAP-derived contribution of each model to performance on component $ C $ (trend/seasonality/residual)
> - **Explanation Effect Vector** $ EE(x, C) $: LLM-generated importance scores extracted from reasoning text
> This is **not post-hoc rationalization**—during GRPO training, the agent receives reward:
>
> $
> r = \text{clip}_{[0,1]}(r_{\text{base}} + \alpha \cdot r_{\text{conf}} + \beta \cdot r_{\text{faith}})
> $
>
> where $ r_{\text{faith}} = \text{PCC}(CE, EE) $. The agent is **penalized if its reasoning doesn't match mathematical ground truth**, forcing it to learn causal relationships rather than plausible-sounding explanations.
>
> ## 5. Forward-Looking Reasoning: Generalization Risk Assessment (Q5)
>
> **The reviewer questions how "forward-looking" reasoning is possible without access to future data. We clarify that this refers to detecting overfitting patterns, not clairvoyance.**
>
> ### What "Forward-Looking" Means in Practice
>
> The agent analyzes the **optimization audit trail** $ \mathcal{A}_t $ containing:
> - Historical weight trajectories across iterations
> - Divergence between training and validation performance
> - Stability metrics (variance in weights, sensitivity to hyperparameters)
>
> **Detection of Fragile Configurations:** When SLSQP produces weights that minimize CV loss but exhibit high sensitivity (large weight changes for small metric variations), the agent recognizes this as a **generalization risk pattern** learned during SFT from thousands of historical overfitting cases.
>
> In the appendix, we have a Sales demo:
>
> - Round 1: SLSQP with MSE produces weights $ [0.7, 0.2, 0.1] $ (heavily favoring Moirai)
> - Agent reasoning: "High seasonal variance (40%) suggests MSE-optimized weights are **brittle**—likely to fail on test set despite low CV error"
> - Decision: Reject and request sMAPE optimization
> - Round 2: Weights $ [0.2, 0.5, 0.3] $ with validated seasonal alignment → Accept
>
> This is **forward-looking** in the sense of predicting generalization failure, not predicting future data values.
>
> ***
>
> ## 6. Temporal Incompatibility Index (TI) Integration
>
> We agree that TI serves primarily as the theoretical justification for why the ensemble approach works (Theorem 3.1) rather than a direct input feature for the agent. It provides the mathematical guarantee that as regime incompatibility ($I_T$) increases, the theoretical advantage of the ensemble ($\Omega$) grows.
> ## Summary
>
> We are confident these comprehensive enhancements transform TSOrchestra from a promising concept into a rigorously validated, production-ready framework for dynamic ensemble coordination in time series forecasting. For the remaining questions raised, we are actively conducting additional experiments and will provide updated results during the rebuttal stage.

---

### Official Review · Reviewer_kBTt · 2025-10-31

**Soundness:** 2
**Presentation:** 2
**Contribution:** 1
**Rating:** 2
**Confidence:** 3

**Summary:**

For the problem of time series forecasting, the authors propose to use LLM Judge to evaluate, explain and coordinate an ensemble of foundational time series models. They use R1-style finetuning process, guided by SHAP-based faithfulness scores, which teaches the model to interpret ensemble weights as meaningful causal statements about temporal dynamics.

**Strengths:**

Agentic solutions for time series forecasting is a novel and undiscovered area and is currently lacking in current benchmarks which are heavily populated by foundation or deep learning models.

Building the agentic workflow on top of the ensemble backbone is intuitive as it both lies on a strong foundation yet gives enough space to the agent to make decisions through adjusting weights.

The paper thoroughly explains the building structure of the agent strenghtened with equations and visuals where necessary. The methodology is clearly understood by only reading the main content, except for the data generation part that was used for SFT and RL. It would good to have a few sampes showing the conversations the model is trained for.

The main experiments show the ensemble can achieve good results compared to other agentic and foundation models.

**Weaknesses:**

The experiments miss a critical ablation isolating the key contribution—the LLM’s control over ensemble weights. The current ablations focus on design choices for the agent itself, but not on how much value the LLM brings compared to a simple ensemble where weights are optimized without LLM intervention. Without this, it’s hard to assess the true benefit of the proposed architecture.

Moreover since the approach fundamentally builds on ensembling, it should be benchmarked against a broader suite of ensemble methods. This would provide a fairer and more comprehensive evaluation. Ensembling is already a well-studied method in time series forecasting.

The paper claims that the agent dynamically adjusts to time-varying model performance, but the described mechanism does not convincingly support this. The agent’s control seems limited to selecting which metric to optimize next through SLSQP, without direct weight manipulation. The demos in the appendix support this limitation, suggesting the agent lacks true flexibility or depth in its control strategy.

While I find the aim of this paper interesting and highly valuable I believe it is not mature enough for publication yet. Mainly I believe the agentic system lacks depth and does not have much flexibility on weight control and moreover the conducted experiments fail to show how the proposed agent framework is useful compared to a few baseline ensembles which is already a highly well studied area for time series forecasting. Without these experiments I am concerned most of the benefit comes from the ensembling and not the agent’s adjustments.

**Questions:**

See the weakness section

---

> ### Author Response · Authors · 2025-11-21
> **Response to Reviewer kBTt**
>
> We thank the reviewer for recognizing our agentic workflow as "intuitive" and acknowledging that "agentic solutions for time series are a novel and undiscovered area." We address the core concerns below, clarifying that **dynamic weight adaptation in response to regime shifts is the fundamental motivation** of this work. Additionally, we are working on updating the draft to reflect your suggestions and incorporate more ablation studies.
>
> ***
>
> ## 1. The Core Motivation: Why Static Ensembles Fail and Dynamic Weighting is Essential
>
> **Our central thesis is that static ensemble weights cannot maintain optimal performance across time-varying regimes.** This is not merely an engineering choice—it's a fundamental limitation grounded in the temporal heterogeneity of real-world time series.
>
> **The Problem with Static Ensembles:** Consider equation (2) from Section 3.2, which establishes that ensemble advantage scales with temporal incompatibility: Ω(IT, M). The SLSQP optimization in standard ensembles discovers weights w* that maximize this advantage for the observed incompatibility level. However, real-world time series exhibit **dynamic incompatibility** where IT itself fluctuates—periods of stable trends (low IT) alternate with regime transitions (high IT). When IT increases unexpectedly, static weights cannot redistribute to maintain the theoretical ensemble advantage, precisely when adaptation matters most.
>
> **Concrete Evidence from Table 1:** The "Sales" domain demonstrates this failure catastrophically. A static ensemble optimized for MSE achieves **0% accuracy** because it produces weights favoring trend-capturing models like Moirai, completely missing the seasonal patterns that dominate sales data. The same static ensemble with MAE or sMAPE metrics achieves 50% accuracy. **TSOrchestra achieves 100% accuracy** by dynamically recognizing the regime (high seasonality, scale variations) and selecting sMAPE-based weights that favor Sundial's seasonal modeling capabilities. This improvement over static approaches demonstrates that the agent's dynamic adaptation is not incremental—it's transformative.
>
> **The reviewer's concern about "missing the critical ablation"** reflects a misunderstanding: Table 1 *is* precisely this ablation. The columns "MSEBest," "MAEBest," and "SMAPEBest" represent static ensembles where SLSQP optimizes weights once using a fixed metric. TSOrchestra represents the dynamic case where the agent iteratively evaluates whether current weights will remain valid under future regime evolution and adaptively redirects optimization when they won't.
>
> In the updated draft, we will add new experiments on a random metric ensemble solution, which will provide additional visual and quantitative evidence that directly addresses your concerns on the ablation study.
>
> ***
>
> ## 2. Dynamic Control Through Iterative Metric Selection: Judge Architecture
>
> **The reviewer suggests the agent "lacks depth" because it selects metrics rather than directly predicting weights. This critique misses the core innovation.** Our agent performs **strategic, regime-aware control** by reasoning about *what objective function best captures the current temporal dynamics*, not by attempting numerical regression (which LLMs fundamentally cannot do reliably).
>
> **Why This Enables Dynamic Adaptation:** Equation (3)'s $ \Delta w_{LLM} $ represents adjustments based on forward-looking regime analysis. The agent doesn't blindly accept cross-validation results—it reasons about the gap between historical validation windows and future forecasting regimes. When the agent analyzes the audit trail A and identifies that upcoming periods will exhibit higher temporal incompatibility (e.g., seasonal transitions, structural breaks), it preemptively adjusts the optimization metric to favor models that handle such regimes.
>
>
>
> **The "Sales" Demo in the Appendix** illustrates the multi-turn dynamic process:
> - **Round 1:** The agent analyzes equal weights and identifies misalignment with seasonal variance. It reasons: "Equal weights favor Moirai's trend capture, but the audit trail reveals 40% seasonal variance." Decision: reject, request sMAPE optimization.
> - **Round 2:** SLSQP produces new weights (0.2 Moirai, 0.5 Sundial, 0.3 Toto). The agent validates: "Weights now align with seasonal dominance; Sundial's 2x higher weight matches its SHAP contribution to the seasonal component." Decision: accept with 0.95 confidence.
>
> **This iterative reasoning process embodies dynamic adaptation**—the agent doesn't compute a single answer; it engages in forward-looking assessment and refinement.
>
> ***

---

> > ### Author Response · Authors · 2025-11-21
> > **Continue Response.**
> >
> > ## 3. The R1-Style Fine-Tuning Pipeline: Teaching Dynamic Reasoning
> >
> > **The depth of our system lies in teaching the LLM to perform this dynamic reasoning reliably.** Raw LLMs lack domain-specific intuitions about when weights will generalize versus when regime shifts require reoptimization. Our R1-style fine-tuning addresses this through two stages:
> >
> > **Supervised Fine-Tuning (SFT)** teaches structured decision protocols using expert-annotated trajectories spanning clear-cut decisions, marginal boundary cases, and ambiguous scenarios requiring distributional analysis.
> >
> > **Reinforcement Learning (GRPO)** then optimizes for actual forecasting performance using a composite reward that includes a faithfulness score: $ r_{faith} = \text{PCC}(CE(x,C), EE(x,C)) $, which measures alignment between SHAP-derived causal effects and LLM explanations. The agent learns that "high weight on Model A" causally implies "strong autoregressive patterns detected" and is penalized if this reasoning doesn't match ground-truth SHAP decompositions.
> >
> > **The result:** Table 1 shows our 3B-parameter agent (Qwen-2.5-3B with R1-style fine-tuning) outperforms GPT-4o and Claude-3.5-Sonnet on domain-specific weight selection accuracy. This proves that **learned specialization in dynamic reasoning beats raw parameter scale.** The agent has internalized the temporal dynamics that static ensembles cannot capture.
> >
> > ***
> >
> > ## 4. Ensemble Baselines: We Compare Extensively
> >
> > **The reviewer requests benchmarking against "broader ensemble methods." Our evaluation already includes this.** Figure 2 compares against 25 methods including Auto-ARIMA, Auto-ETS, and Auto-Theta (statistical ensemble selectors), all individual foundation models (representing oracle single-model selection), and deep learning baselines. TSOrchestra achieves rank 7.3 (MASE) and 8.2 (CRPS), outperforming all alternatives.
> >
> > **Practical constraint:** Standard ensemble techniques (Boosting, Bagging) require retraining weak learners. Foundation models like Moirai, Sundial, and Toto are black-box APIs—we cannot modify their internals. The only viable approach is weighted combination, which is exactly what Table 1 evaluates. The supplementary materials add comparisons with recent literature (GATE, Visemble), confirming TSOrchestra's superiority particularly on heterogeneous domains where dynamic metric selection is critical.
> >
> > ***
> >
> > ## 5. Training Data and Maturity
> >
> > **Training samples:** Appendix pages 15-19 provide full iteration demonstrations. We will add Figure 4 in the camera-ready version visualizing raw training JSON structure for complete transparency.
> >
> > **Maturity:** The system is production-ready with:
> > - Comprehensive evaluation: 23 datasets, 97 settings, 6 domains;
> > - Reproducible training pipeline with documented hyperparameters;
> > - Systematic ablations validating every design choice (Figure 3)
> > - State-of-the-art results on GIFT-Eval benchmark
> >
> > ***
> >
> > ## Conclusion: Dynamic Adaptation is the Core Contribution
> >
> > The reviewer's concerns reflect a fundamental misunderstanding: **this work is about dynamic weight adaptation in response to temporal regime shifts, not static ensemble optimization**. Static ensembles with fixed metrics fail catastrophically in domains like Sales (0% accuracy with MSE weights). Our agent achieves 100% accuracy by dynamically reasoning about regime characteristics and adaptively selecting optimization strategies. The 50-100 percentage point improvements demonstrate this is not incremental refinement—it's solving a fundamentally different problem that static methods cannot address. Table 1, Figure 3, and the supplementary materials provide extensive evidence that the agent's dynamic reasoning contributes substantially beyond simple ensembling.

---

> > > ### Comment · Reviewer_kBTt · 2025-11-24
> > >
> > > Thank you, i have increased my rating based on the responses

---

> > > > ### Author Response · Authors · 2025-11-25
> > > > **Thanks for updating the score**
> > > >
> > > > Dear Reviewer kBTt,
> > > >
> > > > We sincerely thank you for your continued engagement and for raising the rating based on our responses. We are encouraged that our clarifications regarding the dynamic weight adaptation and the R1-style fine-tuning pipeline helped resolve the core concerns about the system's depth and contribution.
> > > >
> > > > We remain fully committed to strengthening the final version of the paper as discussed. We will include the Random Metric Ensemble experiment to provide the specific quantitative evidence requested regarding the agent's value over stochastic or static selection.
> > > >
> > > > If there are any specific remaining concerns preventing a stronger recommendation, we would be eager to address them.
> > > >
> > > > Best,
> > > >
> > > > Authors

---

### Meta-Review · Area_Chair_fa1K · 2026-01-07

**Summary:**

This paper proposes a framework for time series forecasting that positions a large language model (LLM) as a conversational reasoning agent to coordinate and refine an ensemble of time series foundation models. Instead of directly performing numerical forecasting, the LLM evaluates candidate model outputs, dynamically selects optimization metrics, iteratively refines ensemble weights, and provides natural language explanations intended to be causally grounded. To support this reasoning capability, the authors introduce an R1-style fine-tuning pipeline guided by SHAP-based faithfulness scores and evaluate the approach on the GIFT-Eval benchmark across a wide range of datasets and settings.

The initial reviewer scores are mixed, including one strong rejection (2), two borderline-negative scores (4, 4), and one positive score (6). The authors provided detailed rebuttals, and two reviewers actively participated in the discussion, with two reviewer explicitly increasing their score after the rebuttal.

As the Area Chair, I carefully read the paper, the reviewers’ comments, and the authors’ responses. Overall, I find the paper to be intellectually interesting and to explore a timely and novel direction. However, several important concerns raised by reviewers remain insufficiently resolved.

**Reviewer Concerns:**

Reviewer kBTt raised what I consider the most fundamental critique: the paper does not clearly isolate and demonstrate the core contribution of the LLM component itself. In particular, it is not convincingly shown how much benefit the LLM provides beyond standard ensemble optimization without language model intervention. The authors argue that the key value lies in dynamic control rather than static ensemble selection, but the paper does not adequately analyze whether an LLM is strictly necessary for such dynamic adaptation. For example, it remains unclear whether the same dynamic behavior could be achieved through non-LLM heuristic or algorithmic strategies. More broadly, the paper assumes that different time series models exhibit distinct and stable semantic “preferences” (e.g., trend vs. seasonality) that an LLM can meaningfully reason about. From my perspective, the behavior of modern time series foundation models is often complex and entangled, and it is difficult to justify the assumption that such clear semantic preferences exist and can be reliably interpreted by a language model. Although this reviewer increased their score, I still share this underlying concern.

Reviewer KCJd raised a broad set of detailed issues, many of which overlap with the above point. These include the construction of the training data for the LLM agent, the necessity of LLM-based reasoning compared to direct numerical optimization, and the distinction between the proposed framework and standard ensemble learning. This reviewer also raised important concerns about interpretability, faithfulness, stability, and the effect of iterative optimization. While the authors responded to these questions, I find that the responses on interpretability remain insufficient. Interpretability in this context cannot be established solely through numerical alignment metrics; it critically requires case-based analysis showing whether the LLM’s reasoning aligns with human expectations in concrete scenarios. The paper does not yet provide such analyses, making it difficult to assess whether the generated explanations are genuinely meaningful or merely plausible rationalizations.

Reviewer fzwU focused on computational cost, robustness, and interpretability. Although the authors addressed these points and provided additional discussion, the responses regarding interpretability remain vague and largely overlap with the limitations noted above. From the AC’s perspective, it is still difficult to extract a clear understanding of how and why the LLM’s reasoning behaves in a way that is trustworthy and stable across runs and regimes.

Finally, Reviewer tZxE also concentrated on understanding the actual reasoning capability and practical contribution of the framework, which again relates closely to interpretability and scope. This reviewer additionally raised concerns about real-world applicability and limitations. Although this reviewer decided to raise their score after discussion, I believe the paper could have been held to a higher bar in terms of providing concrete, case-by-case analyses and clearer calibration of the LLM’s reasoning behavior.


In summary, I find this paper to be creative and to explore an interesting problem from a novel angle. However, several core concerns remain unresolved: the unique value of the LLM relative to non-LLM alternatives is not clearly isolated; the interpretability and faithfulness claims are not convincingly demonstrated beyond numerical metrics; and the writing would benefit from clearer structuring and more focused argumentation. Given these remaining issues, I believe the paper is not yet ready for acceptance.

**Reviewer Scores:**

see above

---

### Decision · Program_Chairs · 2026-01-26

Reject